# LoPhyDA: Low-Rank Tensor and Physics Gradient Guided Diffusion for Atmospheric Data Assimilation

Danyang Peng [1 2]   Yang Chen [1 2]   Yunlong Zhou [1 2]   Xiao-Tong Yuan [1 2]

## Abstract

Data Assimilation (DA) aims to integrate observations with model forecasts to estimate the state of dynamical systems. Despite the widespread application of diffusion-based assimilation methods, they remain constrained by the high dimensionality of atmospheric states and the reliance on imperfect state-observation mappings. This leaves regions lacking observations spatially unconstrained, leading to severe error accumulation and physical inconsistency.. In this paper, we propose LoPhyDA, a diffusion assimilation algorithm dual-guided by low-rank tensor and physical gradients. By leveraging the low-rank property of meteorological field, we employ tensor completion to exploit spatial continuity and dynamic correlations, reconstructing a globally informative dense field from sparse observations to serve as a global prior. This approach mitigates the information deficit inherent in sparse settings. The framework further incorporates physical constraints into the iterative denoising process, utilizing Partial Differential Equation (PDE) residual gradients to rectify the generative trajectory in real-time. Experimental results demonstrate that LoPhyDA outperforms state-of-the-art generative assimilation models in global weather prediction. It achieves robust and physically consistent assimilation, significantly reducing error accumulation in regions lacking observations.

## 1. Introduction

Accurately estimating the state of Earth systems is critical for reducing initial value errors and improving predictions.

However, such estimations are highly dependent on observational data, which is often insufficient due to real-world constraints. Data Assimilation (DA) achieves state estimation by fusing limited observations with short-term forecasts (background fields) (Lorenc, 1986; Gustafsson et al., 2018), serving as a core technology to enhance forecast accuracy and providing high-quality reanalysis datasets for climate dynamics research (Hersbach et al., 2020). Traditional DA methods used in operational systems include Kalman filtering based on minimum variance estimation and variational methods based on maximum likelihood estimation (Carrassi et al., 2018; Rabier & Liu, 2003). Among these, variational methods estimate the optimal state by minimizing the quadratic cost function of the matching error between the state and observations (Le Dimet & Talagrand, 1986; Asch et al., 2016). Despite their effectiveness, traditional variational methods like 4D-Var are computationally expensive in high-dimensional spaces and often limited by linearization errors when handling non-linear dynamics.

Significant breakthroughs of machine learning in medium-range weather forecasting (Allen et al., 2025; Bi et al., 2023; Lam et al., 2023) has spurred growing interest in Machine Learning-based DA (Yang et al., 2025; Xiao et al., 2025; Andry et al., 2025). Diffusion models offer a novel approach to DA by leveraging their powerful generative capabilities and advantages in conditional modeling (Rozet & Louppe, 2023; Huang et al., 2024; Qu et al., 2024; Sun et al., 2025). Although these methods enhance observation utilization, the sparsity and spatial discontinuity of observations in high-dimensional state spaces leave regions lacking observations without spatial constraints, leading to severe error accumulation and physical inconsistency. While incorporating fundamental physical laws is critical to rectifying physical violations (Luo et al., 2025; Bastek et al., 2025), current physics-guided strategies are only effective in scenarios with dense observational data, and their constraining effect weakens significantly in the face of sparse observations.

Observation reconstruction offers a promising path to address these challenges, as it aims to recover coherent meteorological fields from limited observations (Kadow et al., 2020). While early methods relying on statistical interpolation tools (e.g., multiple imputation MICE (Van Buuren

---

[1]School of Intelligence Science and Technology, Nanjing University, Suzhou, 215163, China. [2]State Key Laboratory for Novel Software Technology, Nanjing University, Nanjing, 210023, China. Correspondence to: Xiao-Tong Yuan <xtyuan@nju.edu.cn>.

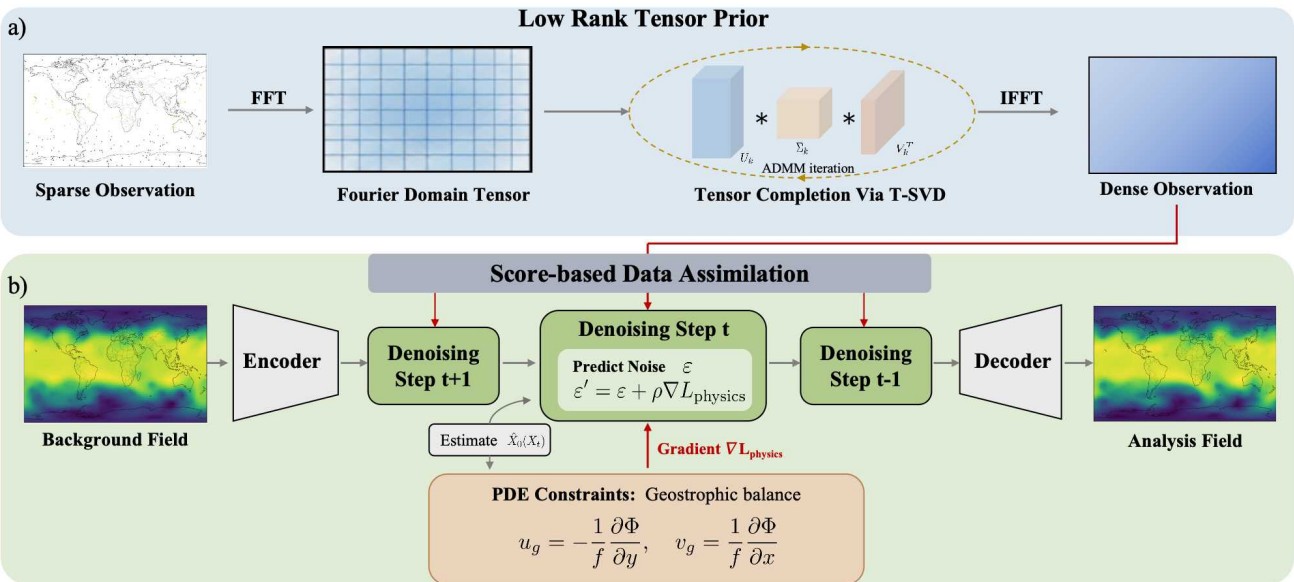

*Figure 1.* Overview of the LoPhyDA framework. **a) Low-rank tensor prior.** Sparse observations are reconstructed via frequency-domain SVD tensor completion to generate a dense global estimate. This reconstruction serves as a global condition for the subsequent diffusion process. **b) Physics gradient guidance.** The reverse diffusion sampling integrates dual constraints: the low-rank prior as a global condition and physical residual gradients. At each step $t$, gradients $\nabla\mathcal{L}_{physics}$ derived from the geostrophic balance are injected into the noise prediction ($\epsilon' = \epsilon + \rho\nabla\mathcal{L}_{physics}$) to enforce dynamical consistency.

& Groothuis-Oudshoorn, 2011)) introduce spurious spatial correlations and oversmooth critical high-frequency structures such as convective boundaries, thereby compromising local observational fidelity. Tensor completion has emerged as numerically efficient techniques for data reconstruction tasks (Chen & Sun, 2021; Liu et al., 2012). By leveraging low-rank prior to capture global structural features, these methods demonstrate robustness and effectiveness even with high missing rates (Luo et al., 2023; Wang et al., 2025b; Zhao et al., 2024; Nie et al., 2024). For instance, Tensor-FLAMINGO (Wang et al., 2025b) employs low-rank strategies to capture global dependencies across genomic loci, enabling the holistic reconstruction of single-cell chromatin structures. Similarly, ImputeFormer (Nie et al., 2024) leverages low-rank prior to model long-range spatiotemporal correlations, extending Transformers to global-scale imputation by balancing structural inductive bias with expressiveness. These cross-domain successes demonstrate that low-rank reconstruction effectively recovers global dependencies in sparse scenarios, offering a theoretical foundation for addressing the lack of spatial constraints.

How to constrain error accumulation in unobserved areas and maintain physical consistency under sparsity remains a critical challenge. To address this, we propose LoPhyDA, a unified diffusion assimilation framework that integrates global prior with physical consistency constraints. We construct a dual-guidance method: First, leveraging the intrinsic low-rank property of meteorological fields, we employ tensor completion to capture spatial continuity and multivariate

dynamical synergy. This reconstructs a globally informative dense initial field from sparse observations, which serves as a global prior to constrain the generative distribution of the diffusion model. Subsequently, we incorporate atmospheric dynamical equations into the reverse diffusion sampling; by computing physical residual gradients and injecting them into the noise prediction steps, we dynamically rectify the iterative denoising path. This dual-guidance strategy ensures that the final analysis field recovers global continuity while adhering to atmospheric dynamical laws. The advantages of our approach are outlined as follows:

- We propose LoPhyDA, which incorporates global prior and local physical constraints into the diffusion model to address the problems of guidance failure and physical distortion under sparse observation scenarios, thereby effectively reducing the uncertainty of state estimation.

- An observation reconstruction method based on low-rank tensors is proposed to reconstruct dense fields from sparse and discontinuous observations, thus enforcing the consistency of global structures.

- A physical gradient-guided method for the inference stage is established. Atmospheric dynamic equations are directly integrated into the diffusion denoising process to correct the generation path in real time, which ensures that the analysis field strictly complies with physical laws in terms of local details.

## 2. Background

**Data assimilation.** The objective of DA is to estimate the atmospheric state $x_t \in \mathbb{R}^n$ at time step t by fusing a background forecast $\hat{x}_t$ with m sparse measurements $y_t = f(x_t^*)$, where $\mathbf{x}_t^*$ denotes the true state. A forecast model $F : \mathbb{R}^n \to \mathbb{R}^n$ provides a prior estimate (background field) $\hat{x}_t = F(\mathbf{x}_{t-1})$. From a probabilistic perspective, DA seeks a state estimate $\mathbf{x}_t$ from the conditional distribution $p(\mathbf{x}_t|\hat{x}_t, \mathbf{y}_t)$ to minimize the discrepancy with the true state $\mathbf{x}_t^*$. To simplify the problem, the observation operator $f$ is typically restricted to a sparse linear operator $\mathbf{y}_t = A\mathbf{x}_t^*$, where $A \in \mathbb{R}^{m \times n}$ is a sparse matrix. This matrix represents point observations, such as measurements of temperature from weather stations or radiosondes.

**Score-based data assimilation (SDA).** The conditional generative capabilities of diffusion models offer a novel paradigm for the modeling of the conditional distribution $p(\mathbf{x} \mid \mathbf{x}_b, \mathbf{y})$ in DA. (Rozet & Louppe, 2023) and (Manshausen et al., 2025) utilize observations as guiding signals during reverse diffusion. However, relying solely on standard conditioning in sparse or noisy observation scenarios often fails to provide sufficient constraints, making reconstruction results susceptible to deviating from physical reality. (Huang et al., 2024) employ a repainting strategy to inject observations into a pre-trained conditional diffusion model. However, this approach relies on linear or simplified observation operators, limiting its applicability to strongly nonlinear processes such as satellite radiance transfer. (Sun et al., 2025) reframes DA as a preference optimization problem, achieving flexible integration of domain knowledge through differentiable reward signals. Overall, these methods fail to overcome the challenges posed by sparse observations: limited data cannot sufficiently constrain global error evolution, nor can it enforce physical guidance in unobserved regions.

**Tensor completion.** Tensor completion has emerged as an effective numerical method for data reconstruction when dealing with high-dimensional missing data. Recently, LowRankOcc (Zhao et al., 2024) achieved accurate completion of outdoor semantic scenes by exploiting the inherent low-rank properties of 3D data. Meanwhile, TiDER (Liu et al., 2023) employed a factorization framework to disentangle multivariate time series into trend and seasonality components, achieving precise modeling of complex dynamics.

## 3. Method

In this section, we propose LoPhyDA. As shown in Figure 1, it constructs a dual-guided diffusion integrating low-rank tensor prior and physical gradients. To address the issue of incomplete observations, we leverage the intrinsic low-rank properties of meteorological fields to reconstruct sparse observations into a globally continuous dense field via tensor completion. And, we introduce a physical gradient constraint based on Partial Differential Equation (PDE), injecting it into the diffusion denoising process to ensure the physical realism of unobserved regions. Finally, we compare LoPhyDA with existing DA methods, analyzing its optimization performance and demonstrating its efficiency.

### 3.1. Sparse-observation guided sampling for diffusion model inference

While score-based diffusion models theoretically overcome the limitations imposed by Gaussian assumptions in traditional DA, in practical scenarios, observational data are typically characterized by sparsity and spatial discontinuity. This physical discontinuity presents a fundamental conflict with the encoder-decoder architecture commonly employed in diffusion models (Huang et al., 2024): the guidance efficacy of sparse observational signals is severely attenuated within the latent space, causing the generation in unobserved regions to degenerate into an unconditional state. To address this challenge, we propose a reconstruction method based on low-rank tensor completion, designed to establish a mapping from discrete observations to continuous, dense observation fields. Unlike coordinate-based models that merely fit positional functions to meteorological grid points, our approach leverages the intrinsic low-rank property of the atmospheric state as prior. This strategy reduces the mapping complexity and imposes a global prior to constrain the solution space in unobserved regions.

**Low-rank tensor prior.** We represent the sparse observation field as a 3-mode tensor $Y \in \mathbb{R}^{I \times J \times N}$, defined on the observed index set $\Omega$. Our objective is to recover the dense tensor $Y'$ from $Y$ by leveraging Tensor Singular Value Decomposition (t-SVD) in the Fourier domain. This process is more than numerical imputation; it is a restoration of the implicit dual-level physical structure. It leverages the low tubal-rank structure along the third dimension to capture dynamical correlations between different meteorological variables (e.g., the geostrophic balance of wind-pressure fields), while utilizing matrix-level low-rankness to capture the spatial continuity of individual variables (i.e., the similarity of atmospheric patterns under comparable circulation backgrounds).

According to tensor completion theory (Lu et al., 2018), the observation tensor $Y_l$ at iteration $l$ is transformed along the tube-wise dimension into the Fourier domain. This transformation indicates that the spectral component $\hat{Y}_{l;i,j,k}$ at longitude $i$ and latitude $j$ aggregates information from all meteorological variables (along the $n$-dimension) at that location. This means that as long as an observed value for any variable exists at that location, the entire corresponding

tube $\hat{Y}_{i,j,:}$ in the Fourier domain can be completed through information integration. The specific Fourier transform is defined as:

$$\hat{Y}_{l;i,j,k} = \sum_{n=1}^{N} Y_{l;i,j,n} \cdot e^{-\frac{2\pi \mathrm{i} k n}{N}}. \tag{1}$$

Given the tensor $\hat{Y}_l$ in the Fourier domain, we apply a soft-thresholding SVD to each frontal slice $\hat{Y}^{(k)}$ (for $k = 1, \ldots, N$) to capture the spatial low-rank structure using information from all grid points. This computes the updated dense tensor slice:

$$\hat{Y}_l' = U_k * \mathcal{S}_\tau(\Sigma_k) * V_k^\top, \tag{2}$$

where $\mathcal{S}_\tau(\cdot)$ denotes the soft-thresholding operator with threshold $\tau = 1/\mu$. Subsequently, the tensor is transformed back via the inverse Fourier transform. The resulting tensor maximally approximates the input, achieving the optimal reconstruction of the observation.

**Optimization objective.** Therefore, the objective function for the low tubal-rank atmospheric field tensor reconstruction is:

$$\min \|Y'\|_{TNN} \quad \text{s.t.} \Omega(Y) = \Omega(Y'), \tag{3}$$

where $Y$ represents the sparse input tensor of observation, $Y'$ represents the dense tensor to be recovered, $\Omega$ represents the set of observed meteorological data in $Y$, and TNN represents the Tensor Nuclear Norm (Qin et al., 2022), which is closely related to the tubal rank of a 3-mode tensor. We employ an efficient Alternating Direction Method of Multipliers (ADMM) (Qin et al., 2022) to solve the optimization problem. Through iterations, the update of $Y'$ is equivalent to solving the following proximal optimization problem:

$$Y'_{l+1} = \arg\min_{Y'} \left( \|Y'\|_{\text{TNN}} + \frac{\mu}{2} \|Y' - (Z^l - \frac{\lambda_2^l}{\mu})\|_F \right), \tag{4}$$

where $Z^l$ represents the intermediate variable derived from the observed data and Lagrange multipliers at the $l$-th ADMM iteration(the detail of ADMM iteration can be found in Appendix B.2).

### 3.2. Physics-informed guidance sampling for diffusion model inference

While the low-rank prior extracts global structures from observations, it inherently relies on data-driven statistical correlations, which often fail to constrain physical violations in unobserved regions. Conversely, existing physics-guided methods typically predicate on dense observational data (Li et al., 2024; Lu et al., 2021), rendering them ineffective when directly applied to sparse scenarios. To address this, we propose a physics gradient guidance DA strategy.

---

**Algorithm 1** LoPhyDA Assimilation Step

---

**Require:** predicted state $\hat{x}$, sparse observation $y$, binary mask $m_h$, localization function Gaspari-Cohn $L$, noise schedule $\{\beta_t\}_{t=1,\cdots,N}$, encoder $E(\cdot)$, decoder $\mathcal{D}(\cdot)$, physics loss $\mathcal{L}_{physics}(\cdot)$, guidance weight $\rho$

1: Output: $x$
2: $y' \leftarrow$ Low-rank tensor prior$(y, m_h)$
3: $z^N \sim \mathcal{N}(0, I)$
4: **for** $t = N, N-1, ..., 1$ *do* **do**
5:    $\alpha_t = \prod_{s=1}^{t}(1 - \beta_s)$
6:    *1. Physics gradient guidance.*
7:    $\epsilon \leftarrow \epsilon_\theta(z_t, t, E(\hat{x}))$   ▷ Conditional noise prediction with background $\hat{x}$
8:    $\hat{z}_0 \leftarrow \frac{1}{\sqrt{\alpha_t}}(z_t - \sqrt{1-\alpha_t}\epsilon)$   ▷ Estimate clean latent (Tweedie's Formula)
9:    $\hat{x}_0 \leftarrow \mathcal{D}(\hat{z}_0)$   ▷ Decode to physical space for PDE check
10:    $g \leftarrow \nabla_{z_t} \mathcal{L}_{physics}(\hat{x}_0)$   ▷ Backpropagate physics gradient
11:    $\hat{\epsilon} \leftarrow \epsilon + \rho\sqrt{1-\alpha_t} \cdot g$   ▷ Inject physics gradient
12:    *2. Predict state*
13:    $\mu_\theta = \frac{1}{\sqrt{1-\beta_t}}(z_t - \frac{\beta_t}{\sqrt{1-\alpha_t}}\hat{\epsilon})$
14:    $\tilde{z}_{t-1} \sim \mathcal{N}(\mu_\theta, \frac{1-\alpha_{t-1}}{1-\alpha_t}\beta_t I)$   ▷ Predict next state
15:    $z_{t-1}^{rec} \sim \mathcal{N}(\sqrt{\alpha_{t-1}}E(y'), (1-\alpha_{t-1})I)$   ▷ Sample noisy reconstructed observation at $t-1$
16:    $z_{t-1} \leftarrow E\left(L \odot D(z_{t-1}^{rec}) + (1-L) \odot D(\tilde{z}_{t-1})\right)$   ▷ Fuse via Gaspari-Cohn
17: **end for**
18: $x \leftarrow \mathcal{D}(z_0)$
19: RETURN $x$

---

Building upon the foundation of low-rank prior, this method leverages PDE gradient information as constraints to achieve precise, physically consistent DA even under sparse observations.

**Physics gradient guidance.** Distinct from physics-guided methods that incorporate weighted constraints into the final training loss (Bastek et al., 2025), our method approaches the problem from the perspective of posterior sampling. Inspired by (Chung et al., 2023), we compute the gradient of the physical residual loss function and inject this gradient into the estimated noise at each denoising step. This correction iteratively optimizes the denoising process, refining the model's output at each stage.

Typically, the noise predicted by the denoising model at time step $t$ is correlated with the model's score in the latent space (Song et al., 2021). Formally, it is expressed as:

$$\epsilon_\phi(z_t, t) = -\sqrt{1-\alpha_t}\nabla_{z_t} \log p(z_t), \tag{5}$$

where $\nabla_{z_t} \log p(z_t)$ denotes the score of the data distribution, and $\alpha_t = \prod_{s=1}^{t}(1 - \beta_s)$ represents the cumulative

*Table 1.* Quantitative comparison and ablation study of LoPhyDA performance under 1% observation condition. Best results are highlighted in bold.

| Model | MAE | MSE | WRMSE | | | | | | |
|---|---|---|---|---|---|---|---|---|---|
| | | | z500 | t850 | t2m | u10 | v10 | u500 | v500 |
| Background | 0.1496 | 0.0565 | 98.3628 | 1.9722 | 5.6729 | 1.2765 | 1.1882 | 2.1649 | 2.3277 |
| SDA | 0.1385 | 0.0401 | 78.7688 | 1.5014 | 5.3589 | 0.9998 | 0.8909 | 2.0228 | 1.4821 |
| DiffDA | 0.1368 | 0.0375 | 75.5332 | 1.3728 | 4.8517 | 0.7659 | 0.7461 | 1.7698 | 1.3842 |
| AlignDA | 0.1267 | 0.0353 | 68.6661 | 1.3076 | 4.6337 | 0.7935 | 0.7245 | 1.6064 | 1.1813 |
| PhyDA | 0.1225 | 0.0347 | 71.6741 | **1.1033** | 4.7928 | 0.8062 | 0.7196 | **1.2435** | 1.4766 |
| **LoPhyDA** | **0.1183** (↓3.55%) | **0.0336** (↓3.27%) | **58.8376** | 1.1217 | **4.4293** | **0.7178** | **0.6811** | 1.4919 | **1.0229** |

noise schedule parameter at step $t$, derived from the predefined variance schedule $\beta_s$. However, we need to consider not only the data prior but also incorporate the physical constraint $g$ during the sampling process of the diffusion model.

In our work, $F(\cdot)$ denotes the physical residual operator. To enforce physical constraints, we set the target residual $g = \mathbf{0}$ as condition. Leveraging Bayesian inference, we introduce the gradient $\nabla_{z_t} \log p(g \mid z_t)$ to rectify the sampling trajectory; this term approximates the direction minimizing the residual norm $\|g - F(\mathcal{D}(\hat{z}_0(z_t)))\|_2^2$, thereby guiding the generated state toward physical consistency. Consequently, the score of the denoising model at time step $t$ becomes $\nabla_{z_t} \log p(z_t \mid g)$. Since this term is intractable, we derive it using the known $\nabla_{z_t} \log p(z_t)$. Based on Bayes' theorem, we can express this as:

$$\nabla_{z_t} \log p(z_t \mid g) = \nabla_{z_t} \log p(z_t) + \nabla_{z_t} \log p(g \mid z_t). \tag{6}$$

Thus, given that the first term is known, the problem reduces to calculating the physical likelihood gradient $\nabla_{z_t} \log p(g \mid z_t)$. Inspired by (Chung et al., 2023), we leverage Tweedie's formula and the pre-trained decoder $\mathcal{D}$ to derive the following approximation:

$$\begin{aligned} \nabla_{z_t} \log p(g \mid z_t) &\simeq \nabla_{z_t} \log p\left(g \mid \mathcal{D}(\hat{z}_0(z_t))\right) \\ &\simeq -\rho \nabla_{z_t} \|g - F\left(\mathcal{D}(\hat{z}_0(z_t))\right)\|_2^2, \end{aligned} \tag{7}$$

where $\nabla_{z_t} \|g - F\left(\mathcal{D}(\hat{z}_0(z_t))\right)\|_2^2$ is also denoted as $\nabla_{z_t} \mathcal{L}_{physics}$. Therefore, we express the noise prediction adjusted by the constraint $g$ as:

$$\begin{aligned} \epsilon'_\phi &= \epsilon_\phi(z_t, t) + \rho\sqrt{1 - \alpha_t} \nabla_{z_t} \|g - F\left(\mathcal{D}(\hat{z}_0(z_t))\right)\|_2^2 \\ &= \epsilon_\phi(z_t, t) + \rho\sqrt{1 - \alpha_t} \nabla_{z_t} \mathcal{L}_{physics}, \end{aligned} \tag{8}$$

where $\epsilon'_\phi$ denotes the adjusted noise, obtained by adding the gradient of the guidance loss $\nabla_{z_t} \mathcal{L}_{physics}$ to the noise predicted by the original denoising model. And, the hyperparameter $\rho$ scales the magnitude of the injected physical gradient.

**Physics constraints.** To ensure that the assimilated analysis field possesses physical consistency, we introduce a physical loss term, $\mathcal{L}_{\text{physics}}$, designed to enforce the geostrophic balance constraint on the decoded state $\hat{x}_0$. Let $U, V$, and $\Phi$ denote the zonal wind, meridional wind, and geopotential components extracted from $\hat{x}_0$, respectively. The theoretical geostrophic winds derived from the geopotential $\Phi$ are given by:

$$u_g = -\frac{1}{f}\frac{\partial \Phi}{\partial y}, \quad v_g = \frac{1}{f}\frac{\partial \Phi}{\partial x}, \tag{9}$$

where $f$ is the Coriolis parameter. Accordingly, the physical loss is defined as the sum of squared residuals quantifying the deviation of the estimated wind field $(U, V)$ from the geostrophic equilibrium:

$$\mathcal{L}_{\text{physics}}(\hat{x}_0) = \|U - u_g(\Phi)\|_2^2 + \|V - v_g(\Phi)\|_2^2. \tag{10}$$

To formulate the assimilation process, we fuse the predicted state $\tilde{z}_t$ with the noisy reconstructed observation $z_t^{rec}$ using the Gaspari-Cohn localization function $L$ (Gaspari & Cohn, 1999) (Appendix B.3). Specifically, the observation is sampled as $z_t^{rec} \sim \mathcal{N}(\sqrt{\alpha_t}E(y'), (1 - \alpha_t)I)$, yielding the updated state $z_t = E(L \odot \mathcal{D}(z_t^{rec}) + (1 - L) \odot \mathcal{D}(\tilde{z}_t))$, where $E(\cdot)$ denotes the VAE encoder. The overall algorithm of applying the denoising diffusion model for data assimilation is presented in Algorithm 1.

## 4. Experiment

### 4.1. Experimental settings and implementation

**Datasets.** Experiments were conducted on the ERA5 reanalysis dataset (Hersbach et al., 2020), a global atmospheric data product maintained by the European Centre for Medium-Range Weather Forecasts. In this study, a total of 69 key variables were selected for experimentation, comprising five categories of upper-air variables—Geopotential $(Z)$, Temperature $(T)$, Specific Humidity $(Q)$, Zonal Wind Component $(U)$, and Meridional Wind Component $(V)$—each available across 13 standard pressure levels (50, 100, 150, 200, 250, 300, 400, 500, 600, 700, 850, 925, and 1000 hPa).

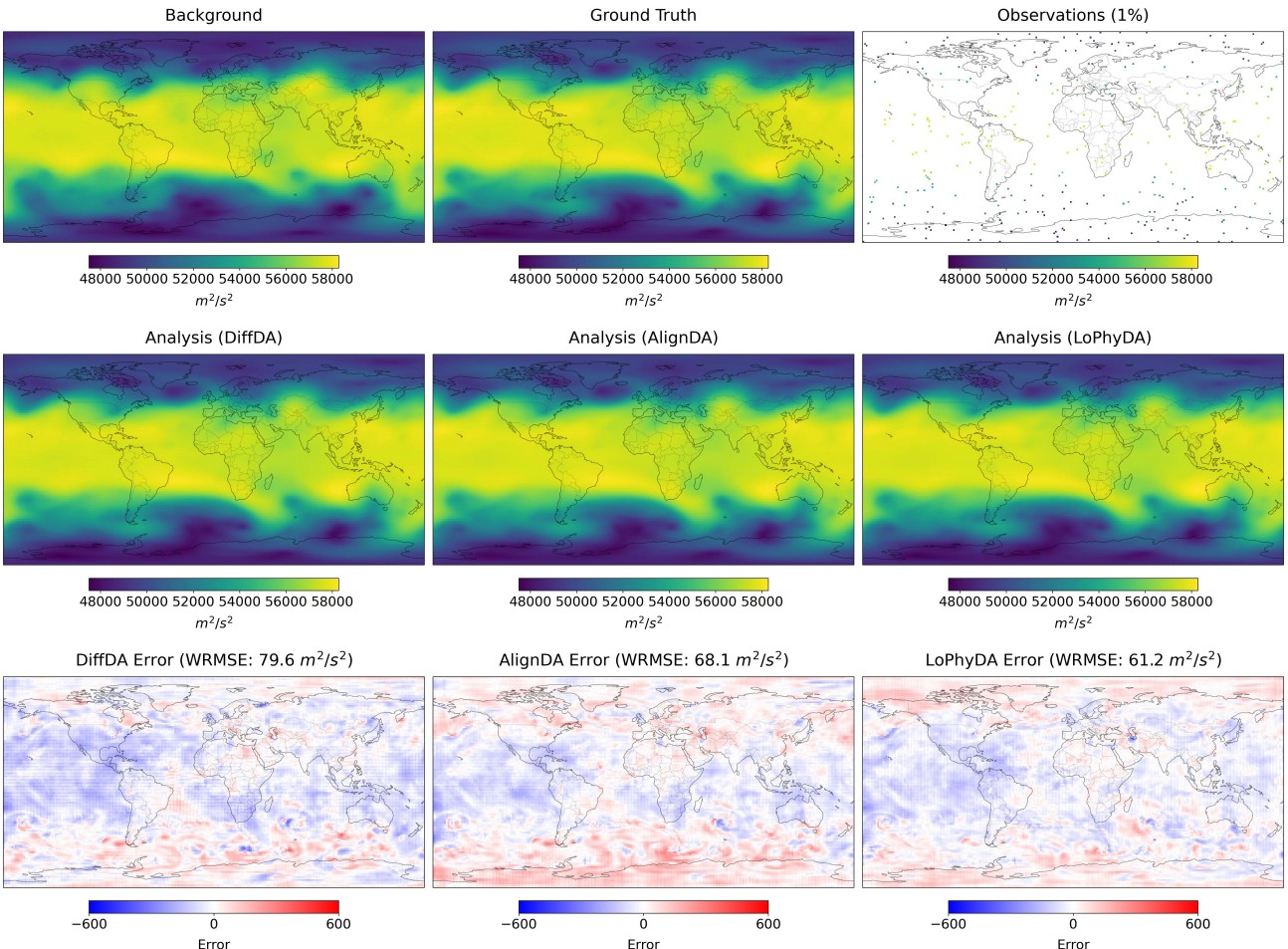

*Figure 2.* Comparative visualization of Z500 analysis fields across assimilation methods under 1% simulated observation (valid at 2019-01-06 00:00 UTC). The top row present the ERA5 ground truth, initial background field, and sparse observations. The middle and bottom rows contrast the assimilation results and corresponding absolute errors for DiffDA, AlignDA, and LoPhyDA. In the error maps, the noticeably reduced magnitude and lighter hues achieved by LoPhyDA highlight its superior capability in suppressing deviations within unobserved regions compared to alternative approaches.

the 13 sub-variables across various vertical levels are denoted using abbreviations of their short names and pressure levels (e.g., z500 represents geopotential height at 500 hPa). Additionally, four surface variables were selected: 10m Zonal Wind ($U10$), 10m Meridional Wind ($V10$), 2m Temperature ($T2M$), and Mean Sea Level Pressure ($MSLP$). Following the convention of the paper (Sun et al., 2025), we utilize the ERA5 dataset from 1979 to 2015 to train a 6-hour forecast model, with a spatial resolution of $1.40625°$ ($128 \times 256$ grid).

**Observation settings.** We conduct experiments using two types of observations. For simulated observations, we generate data based on ERA5 reanalysis by adopting the sparse grid column sampling method (Huang et al., 2024). Specifically, observation instances are randomly selected from the global grid to mimic the sparse distribution characteristic of real-world scenarios. For real-world observations, we utilize

the GDAS prepbufr dataset(Rodell et al., 2004), which aggregates measurements from diverse observing systems and instruments. These observations often fall off-grid and are typically sparse across vertical levels, closely resembling operational assimilation scenarios.

**Baseline.** We compare LoPhyDA against four baselines: (1) Background, which conditions the reverse diffusion process solely on the background field. (2) SDA (Rozet & Louppe, 2023), which employs observations as guidance signals during reverse diffusion. (3) DiffDA (Huang et al., 2024), which repaints observations during sampling and integrates them with the background field to condition the diffusion. (4) AlignDA (Sun et al., 2025), which integrates prior knowledge via soft constraints during diffusion assimilation. (5) PhyDA (Wang et al., 2025a), which introduces physical constraints for DA. To ensure a fair comparison and architectural consistency, we implemented the latent-space

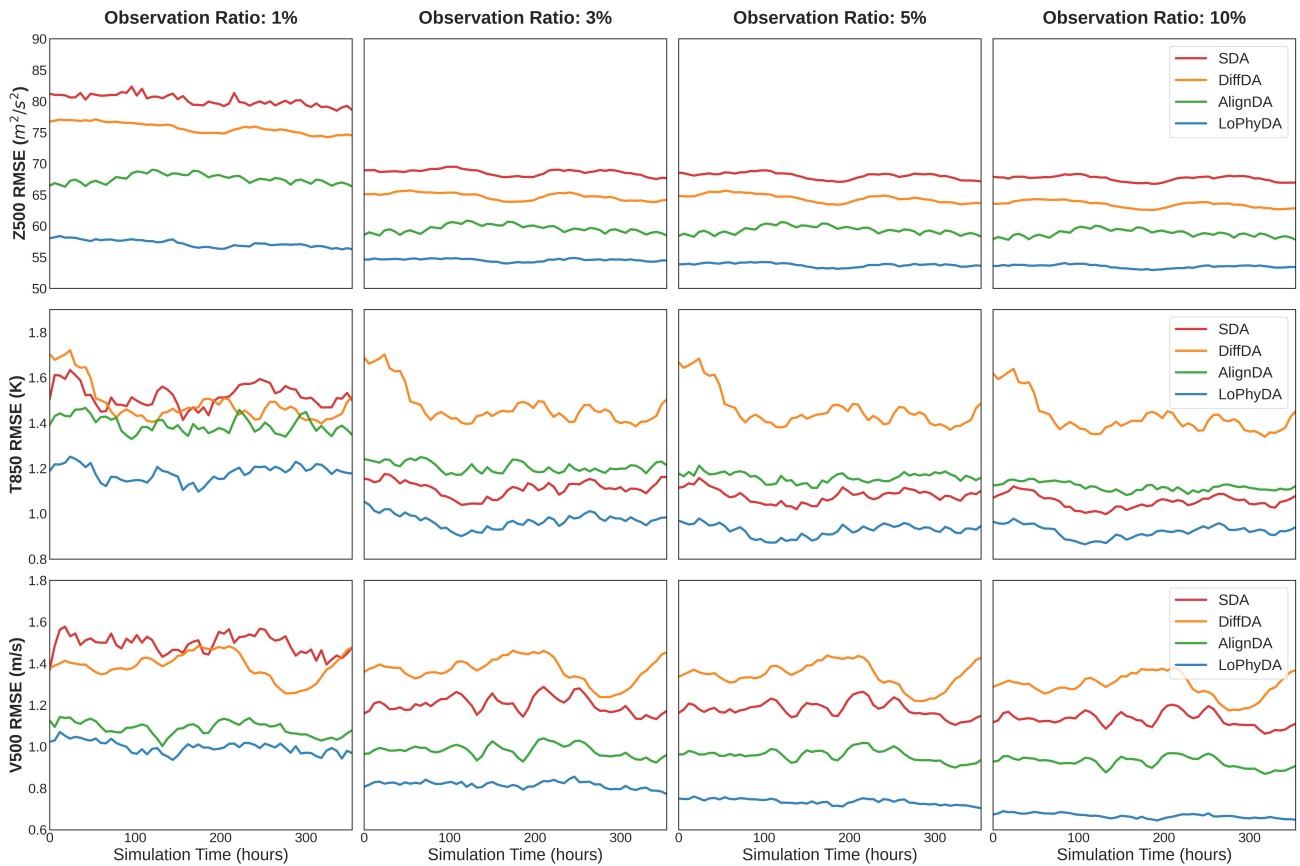

*Figure 3.* Results for fixed observation positions (15 days). The system is simulated in an auto-regressive manner for 15 days, starting from January 1, 2019. Following AlignDA, we demonstrate the RMSEs of the analysis field for three variables (z500, t850, v500) in three different rows.

versions of all diffusion-based baselines (SDA, DiffDA, and AlignDA). Specifically, they share the same pre-trained VAEformer backbone and operate within the same latent space as LoPhyDA.

**Model training and evaluation.** The model was trained for 20 epochs using the AdamW optimizer with a learning rate of $1 \times 10^{-4}$. We used ERA5 data from 1979 to 2015 as the training set, data from 2016 to 2017 as the validation set, and ERA5 data from 2018 as the test set. Training was conducted on four NVIDIA RTX 4090 GPUs with a global batch size of 8 and took approximately 3 days. Unless otherwise specified, all models were trained under the same data split and optimization settings to ensure a fair comparison.

After training, we evaluate model performance by computing the overall mean squared error (MSE), mean absolute error (MAE), and latitude-weighted root mean square error (WRMSE). WRMSE is a widely used statistical metric in geospatial analysis and atmospheric science. Given the prediction $\hat{x}_{h,w,c}$ and the corresponding ground truth $x_{h,w,c}$ at latitude index $h$, longitude index $w$, and channel $c$, WRMSE

is defined as:

$$
\mathrm{WRMSE}(c) = \left[ \frac{1}{HW} \sum_{h,w} H \cdot \left( \frac{\cos \alpha_{h,w}}{\sum_{h'=1}^{H} \cos \alpha_{h',w}} \right) \right.
$$
$$
\left. \times \left( x_{h,w,c} - \hat{x}_{h,w,c} \right)^2 \right]^{1/2},
\tag{11}
$$

where $H$ and $W$ denote the number of grid points in the longitudinal and latitudinal directions, respectively; $\alpha_{h,w}$ represents the latitude of the grid point $(h, w)$.

### 4.2. Single-step data sssimilation

We evaluate LoPhyDA's effectiveness on a single-step assimilation task. The background field was generated using the FengWu model (Chen et al., 2023), initialized from the ERA5 reanalysis on December 30, 2018, and evolved over 48 hours.

Table 1 presents a quantitative comparison between Lo-PhyDA and other assimilation methods under the 1% observation condition. To ensure statistical robustness, all

*Table 2.* Ablation study of LoPhyDA: Effectiveness of Low-Rank Tensor constraint and physical guidance under 1% observation condition. Best results are highlighted in bold.

| Model | MAE | MSE | WRMSE | | | | | | |
| --- | --- | --- | --- | --- | --- | --- | --- | --- | --- |
| | | | z500 | t850 | t2m | u10 | v10 | u500 | v500 |
| LDA | 0.1386 | 0.0415 | 82.3469 | 1.4726 | 5.3712 | 0.9765 | 0.8738 | 1.9743 | 1.5238 |
| LDA-Phy | 0.1319 | 0.0388 | 75.4131 | 1.4964 | 5.2932 | 0.9236 | 0.8225 | 1.7681 | 1.3059 |
| LDA-LoR | 0.1235 | 0.0365 | 65.3315 | 1.2312 | 4.6135 | 0.7203 | 0.7367 | 1.6645 | 1.2158 |
| **LoPhyDA** | **0.1183** | **0.0336** | **58.8376** | **1.1217** | **4.4293** | **0.7178** | **0.6811** | **1.4919** | **1.0229** |

reported metrics (MAE, MSE, and WRMSE) are averaged over the entire 2019 test set, reflecting the model's consistent performance across varying atmospheric states. Our method consistently achieves the lowest mean values across all evaluation metrics, demonstrating improved performance in assimilation accuracy. In sparse observation scenarios, DiffDA enhances observation guidance via a repainting algorithm but imposes insufficient error constraints on unobserved regions. In contrast, our method achieves more accurate predictions through sparse observation reconstruction and physical gradient constraints. Compared to the best baseline PhyDA, LoPhyDA reduces the MSE by approximately 3.55%. This indicates that introducing the low-rank reconstructed observation field as a condition for the posterior distribution yields superior assimilation effects compared to relying on reward-guided distribution alignment.

Figure 2 presents a spatial evaluation of assimilation quality using Z500. The visualization reveals that DiffDA exhibits distinct artifacts and significant error fluctuations (indicated by deep red/blue regions) in unobserved areas. In contrast, LoPhyDA significantly suppresses these large deviations, with its error map displaying extensive areas of near-zero values (lighter regions). This improvement is particularly pronounced in high-latitude regions characterized by complex dynamics. These results suggest that LoPhyDA effectively recovers observations containing global information via low-rank reconstruction, establishing a robust prior for diffusion assimilation. Coupled with physical gradient constraints, the method not only corrects numerical deviations in unobserved regions but also ensures the spatial continuity and physical plausibility of the variables.

### 4.3. Cyclic assimilation and forecasting

To evaluate the long-term performance of LoPhyDA in a quasi-operational setting, we conducted a 15-day cyclic assimilation and forecast experiment starting from January 1, 2019, at 00:00 UTC. The experiment employed a 6-hour assimilation cycle, where the analysis field from the previous cycle was used to initialize the subsequent forecast. We implemented two sampling strategies for the simulated observation: fixed-location sampling and unfixed sampling.

*Table 3.* Comparison of different observation field reconstruction methods.

| Method | MAE | MSE |
| --- | --- | --- |
| Kriging Interpolation | 0.1328 | 0.0391 |
| Learnable Encoder | 0.1237 | 0.0352 |
| **Low-Rank** | **0.1183** | **0.0336** |

In the fixed sampling experiments, where observation locations remain constant over time, we tested four observation ratios (1%, 3%, 5%, and 10%) to maintain consistency with AlignDA. Assimilation accuracy was evaluated using the Root Mean Square Error (RMSE) of the analysis state. As shown in Figure 3, LoPhyDA consistently and stably outperforms the three baseline methods across all three demonstrated variables (results for other variables are provided in the Appendix E). This indicates that our method effectively reduces errors and systematic biases within the cyclic prediction system. Furthermore, in non-fixed sampling experiments where observation locations vary over time (see Appendix C.2), LoPhyDA maintains its performance advantage over DiffDA and AlignDA, yielding results consistent with the fixed sampling scenarios.

### 4.4. Ablation study

We conducted an ablation study on low-rank prior and physical gradients, with Table 2 presenting three configurations evaluated under 1% observation: the baseline Latent Diffusion Assimilation (LDA) without low-rank prior and physics gradient, LDA-Phy incorporating only physical constraints, and LDA-LoR employing only low-rank prior.

LDA-Phy primarily utilizes the geostrophic balance equation as physical guidance , reducing the MSE from 0.0415 to 0.0388 and validating the precise constraining effect of physical gradients on the solution space. This advantage is particularly evident in wind field variables such as U10 and V10, where it effectively corrects physical distortions in unobserved regions. In contrast, LDA-LoR, which introduces the low-rank prior, demonstrates more significant gains, further reducing the MSE to 0.0365. For the Z500 variable, LDA-LoR achieves an error of 65.3315 , outperforming the

*Table 4.* Assimilation performance on real-world observation across various methods. Best results are highlighted in bold.

| Model | MAE | MSE | WRMSE | | | | | | |
|---|---|---|---|---|---|---|---|---|---|
| | | | z500 | t850 | t2m | u10 | v10 | u500 | v500 |
| Background | 0.1440 | 0.0571 | 99.8706 | 1.9692 | 5.8989 | 1.2588 | 1.2406 | 2.0761 | 2.4066 |
| DiffDA | 0.1275 | 0.0428 | 78.8742 | 1.4428 | 4.7613 | 0.8935 | 0.7321 | 1.7698 | 1.4950 |
| AlignDA | 0.1241 | 0.0413 | 73.2120 | 1.3709 | 4.5811 | 0.8164 | 0.7069 | 1.6964 | 1.3748 |
| **LoPhyDA** | **0.1206** (↓2.82%) | **0.0385** (↓6.78%) | **65.4817** | **1.0376** | **4.3791** | **0.6874** | **0.7005** | **1.4075** | **1.2639** |

baseline models AlignDA (68.6661) and DiffDA (75.5332). This demonstrates that reconstructing observation fields by leveraging the low-rank properties of meteorological data effectively captures global spatial dependencies and provides a richer informational prior for assimilation. Ultimately, the LoPhyDA model achieves an optimal Z500 error of 58.8376 by synergizing the low-rank prior with physical gradient constraints, effectively unifying high-precision observation fitting with fundamental atmospheric dynamics.

To further validate the effectiveness of the low-rank prior in reconstructing sparse observation fields, we compare it with Kriging interpolation and the learnable encoder used in PhyDA, as shown in Table 3. The low-rank method achieves the best performance, with an MAE of 0.1183 and an MSE of 0.0336. Compared with Kriging interpolation, it reduces MAE from 0.1328 to 0.1183 and MSE from 0.0391 to 0.0336, corresponding to relative improvements of 10.9% and 14.1%, respectively. Compared with the learnable encoder, the low-rank method further reduces MAE by 4.4% and MSE by 4.5%. These results indicate that the low-rank prior can reconstruct a more accurate and dense observation field. In contrast, the learnable encoder requires repeated training for different observation densities, which limits its generalizability; while Kriging interpolation is training-free, it struggles to preserve global structural information and tends to produce localized fitting.

### 4.5. Real-world observations

To evaluate our framework under real-world conditions, we utilize the Global Data Assimilation System (GDAS) Prep-bufr dataset , which aggregates measurements from diverse observation systems. These observations typically exhibit off-grid characteristics and vertical sparsity, closely reflecting operational DA scenarios. Using 48-hour background fields at 00:00 UTC daily throughout 2017, we performed comparative experiments. As shown in Table 4, LoPhyDA consistently outperforms both DiffDA and AlignDA. Notably, compared to the second-best performing AlignDA, LoPhyDA reduces the MAE by 2.82% and the MSE by 6.78%. These results demonstrate superior accuracy and robustness in handling real-world observations, highlighting LoPhyDA's potential for DA and weather forecasting.

## 5. Conclusion and Discussion

**Conclusion.** This paper presents LoPhyDA, a novel data assimilation framework that addresses the challenges of high-dimensional sparse observations through a dual-guidance diffusion paradigm. Unlike traditional end-to-end training methods, LoPhyDA leverages the intrinsic low-rank properties of meteorological fields to reconstruct global, continuous prior from discrete measurements via iterative numerical optimization.

Notably, LoPhyDA establishes a retraining-free, inference-stage adaptation mechanism. By decoupling observation reconstruction from the generative backbone and enforcing physical consistency through gradient injection, the framework adaptively handles diverse observation densities and physical constraints without additional training costs. Experimental results on global weather demonstrate that LoPhyDA significantly reduces artifacts in unobserved regions and minimizes physical violations. hese insights offer an efficient path for next-generation physically consistent DA systems and potential meteorological extensions like forecast post-processing.

**Limitations and future work.** LoPhyDA achieves sparse observation reconstruction and physics-guided data assimilation within the diffusion model framework. However, the iterative process of diffusion models leads to high computational cost for online assimilation, making it difficult to meet real-time requirements. In the future, we will introduce the flow matching framework, which enables direct distribution mapping to avoid multi-step iterations and improve the speed of online assimilation. In addition, the regional applicability of current physical constraints remains limited. LoPhyDA mainly adopts the geostrophic balance equation as physical guidance, but this equation is an idealized dynamical approximation and does not fully hold in equatorial and low-latitude regions. Therefore, in future work, we plan to incorporate more region-adaptive physical equations and design specialized modules to disable or weaken the geostrophic balance constraint in low-latitude regions, instead relying on other physical priors to ensure the physical consistency of assimilation results.

## Acknowledgments

This work was supported by the Natural Science Foundation of China (No. U21B2049); the Special Fund for Key Program of Science and Technology of Jiangsu Province (No. BG2024042); the Gusu Leading Talents Program for Innovation and Entrepreneurship (No. ZXL2025323); the Fundamental and Interdisciplinary Disciplines Breakthrough Plan of the Ministry of Education of China under Grant (No. JYB2025XDXM118); and the "111 Center" (No. B26023). We thank the ICML reviewers for their valuable feedback and suggestions.

## Impact Statement

This paper aims to advance the application of machine learning in global weather data assimilation and forecasting research. Our work has many potential societal consequences, none of which we feel must be specifically highlighted here.

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

## A. Algorithm analysis

**Distinctions in physical guidance.** While both PhyDA and LoPhyDA incorporate physical guidance within the diffusion denoising process, they operate at fundamentally different levels. PhyDA introduces physical constraints as a regularization term in the objective function during the training phase to update the denoising network parameters. Consequently, the physical constraints are hardcoded into the model weights, resulting in high retraining costs. In contrast, LoPhyDA does not alter the pre-trained network. Instead, during the inference phase, it directly injects the physical residual gradients into each step of the reverse denoising process. Its target is the sampling trajectory of the current sample rather than the model parameters. Therefore, LoPhyDA is a training-free method and offers the advantage of dynamically adjusting the guidance strength based on observation density and time steps.

$$\epsilon'_\phi = \epsilon_\phi(z_t, t) + \rho\sqrt{1 - \bar{\alpha}_t}\,\nabla_{z_t}\mathcal{L}_{\text{physics}}. \tag{12}$$

As illustrated in this formula, $\epsilon_\phi$ is the noise predicted by the pre-trained denoising model. By adding the physical gradient to this noise, we obtain the adjusted, physics-informed noise $\epsilon'_\phi$.

**Distinctions in handling sparse observations.** PhyDA forces the neural network to directly learn the coupling relationships between variables from sparse observations. When observations are extremely sparse, it is prone to establishing unreasonable correlations between different variables. LoPhyDA, however, first decouples the variable relationships in the Fourier domain, and then achieves cross-variable joint reconstruction through low-rank constraints along the variable dimension. This strategy ensures the relative physical independence of each variable's reconstruction while simultaneously capturing cross-variable connections through a shared low-rank structure, effectively realizing a "decouple first, jointly reconstruct later under structural constraints" method.

**Distinctions from Appa.** Appa designs a spherical geometry image encoder capable of processing global weather data, which differs from conventional planar grid encoders. Its primary function is to achieve the compression of high-dimensional data while preserving spherical features. This objective is fundamentally different from that of LoPhyDA, which aims to reconstruct highly sparse and discrete observation data into observation fields with dense information.

## B. Additional implementation details

### B.1. The VAE training

In this study, we adopt the VAEformer architecture proposed by (Han et al., 2024), whose core function is to realize an efficient mapping from high-dimensional atmospheric fields to a low-dimensional latent space. This model incorporates the window attention mechanism proposed by (Liu et al., 2021), enabling it to accurately and efficiently capture the core characteristics of atmospheric circulation systems. The model parameters are configured as follows: both the size and stride of patch embedding are set to $(4, 4)$, the embedding dimension is configured as $1024$, and the overall network structure is constructed by stacking $24$ Transformer blocks integrated with the window attention mechanism layer by layer. During the model training phase, the AdamW optimizer is employed to drive parameter updates, with a total of $50$ training epochs completed. The batch size is fixed at $32$ throughout the training process, and the initial learning rate is set to $10^{-4}$. The ERA5 reanalysis dataset is selected as the training dataset, covering the time span from 1979 to 2015. To comprehensively evaluate the generalization performance of the model, the data from 2018 is used as the validation set. Experimental results demonstrate that the trained VAE model achieves a mean square error (MSE) of $0.0073$ and a mean absolute error (MAE) of $0.0518$ on the validation set, which fully verifies that the model possesses excellent fitting accuracy.

### B.2. Low-rank tensor prior

LoPhyDA adopts an efficient Alternating Direction Method of Multipliers (ADMM) strategy to enforce the low-rank tensor prior. To facilitate the ADMM solution, the objective function is reformulated as follows:

$$\min_{Y', Z} \|Y'\|_{TNN}, \quad s.t.\ d = A \times Vec(Z), \quad Y' = Z, \tag{13}$$

where $\mathbf{d} = \mathbf{A} \times \text{Vec}(\mathbf{Y})$. Here, $\mathbf{A}$ is a binary matrix taking values of 1 on the set $\Omega$ and 0 otherwise. $\text{Vec}(\cdot)$ denotes the vectorization process of a tensor. The Lagrangian function can be formulated as:

$$L(Y', Z, \lambda_1, \lambda_2) = \|Y'\|_* + \langle\lambda_1, d - A \times Vec(Z)\rangle + \langle\lambda_2, Y' - Z\rangle$$
$$+ \frac{\mu}{2}|d - A \times Vec(Z)|^2 + \frac{\mu}{2}|Y' - Z|^2. \tag{14}$$

Accordingly, $Y'$, $Z$, and the Lagrangian multipliers $\lambda_1$, $\lambda_2$ can be updated through iterations as:

$$
\begin{aligned}
Y'_{l+1} &= \text{argmin}_{Y'} \|Y'\|_* + \frac{\mu}{2} \|Y' - Z^l + \lambda_2^l/\mu\|_F \\
Z^{l+1} &= (A^T A + I)^{-1}(A^T \lambda_1^l/\mu + Vec(\lambda_2^l)/\mu + A^T d + Vec(Y'_{l+1})) \\
\lambda_1^{l+1} &= \lambda_1^l + \mu(A \times Vec(Z^{l+1}) - d) \\
\lambda_2^{l+1} &= \lambda_2^l + \mu(Y'_{l+1} - Z^{l+1}).
\end{aligned}
\tag{15}
$$

In each iteration, the update of $Y'$ involves an optimization step, which is solved by the t-SVD method. Through iterations, the updating scheme of $Y'$ is equivalent to solving the following optimization problem:

$$
Y'_{l+1} = \arg\min_{Y'} \left( \|Y'\|_{\text{TNN}} + \frac{\mu}{2} \|Y' - (Z^l - \frac{\lambda_2^l}{\mu})\|_F \right).
\tag{16}
$$

This optimization problem can be solved using the t-SVD algorithm (Lu et al., 2018). The SVD captures the spatial low-rank structure of the variable by utilizing information across all grid points, and the tensor is transformed back via the inverse Fourier transform. The resulting tensor maximally approximates the input, ultimately achieving the optimal reconstruction of the original observations.

### B.3. Localization function

During the reverse diffusion sampling, the conditional noise prediction relies on the background field $E(\hat{x})$. To integrate the observational data, we apply the Gaspari-Cohn (G-C) localization function $L$ at each timestep to fuse the decoded noisy reconstructed observation $\mathcal{D}(z_{t-1}^{rec})$ with the decoded predicted state $\mathcal{D}(\tilde{z}_{t-1})$. $L$ is a fifth-order piecewise rational function with compact support (Gaspari & Cohn, 1999), used to blend the t-SVD reconstructed field $y'$ with the background field $\hat{x}$. To account for the spatial uncertainty of the reconstruction, the localization weight $L(d; c)$ is calculated based on the distance $d$ to the nearest original observation site:

$$
L(d; c) = \begin{cases}
-\frac{1}{4}(\frac{d}{c})^5 + \frac{1}{2}(\frac{d}{c})^4 + \frac{5}{8}(\frac{d}{c})^3 - \frac{5}{3}(\frac{d}{c})^2 + 1, & 0 \le d \le c \\
\frac{1}{12}(\frac{d}{c})^5 - \frac{1}{2}(\frac{d}{c})^4 + \frac{5}{8}(\frac{d}{c})^3 + \frac{5}{3}(\frac{d}{c})^2 - 5(\frac{d}{c}) + 4 - \frac{2}{3}(\frac{d}{c})^{-1}, & c < d \le 2c \\
0, & d > 2c
\end{cases}
\tag{17}
$$

where $c$ is localization parameter that controls the spatial influence range. The function smoothly decays with increasing distance $d$ and vanishes completely beyond the cut-off distance of $2c$. This function ensures that $z_{cond}$ provides high-confidence spatial constraints, effectively eliminating spurious long-range correlations while maintaining physical continuity.

## C. Additional experimental analysis

### C.1. Observation density analysis

We further investigated the impact of different observation densities (1%, 3%, 5%, and 10%) on the effectiveness of our proposed framework, as shown in Table 5. A key finding is that LoPhyDA exhibits continuous and significant accuracy improvements across all evaluation metrics as observational information becomes richer. Specifically, the MSE steadily decreased from 0.0336 at 1% observation to 0.0289 at 10%, achieving a relative improvement of approximately 14.0%.

*Table 5*. Assimilation accuracy gains with different observation densities.

| | MAE | MSE | WRMSE | | | | | | | |
| --- | --- | --- | --- | --- | --- | --- | --- | --- | --- | --- |
| | | | z500 | t850 | t2m | u10 | v10 | u500 | v500 | q700 |
| 1% observation | 0.1183 | 0.0336 | 58.8376 | 1.1217 | 4.4293 | 0.7178 | 0.6811 | 1.4919 | 1.0229 | 0.0006 |
| 3% observation | 0.1121 | 0.0311 | 57.9902 | 1.1018 | 4.2437 | 0.5534 | 0.6358 | 1.2751 | 0.9567 | 0.0006 |
| 5% observation | 0.1094 | 0.0302 | 57.5785 | 1.0826 | 4.2287 | 0.5079 | 0.6261 | 1.2031 | 0.8768 | 0.0005 |
| 10% observation | **0.1054** | **0.0289** | **56.9307** | **1.0245** | **4.2053** | **0.4381** | **0.6126** | **1.0971** | **0.7536** | **0.0005** |

## C.2. Computational cost

The training of our background conditional diffusion model takes 2.5 days on 4 Nvidia RTX-4090 GPUs. When performing DA, our LoPhyDA technique consume about 15 seconds per sample on an Nvidia RTX-4090 GPU.

*Table 6.* Computational complexity breakdown of LoPhyDA during inference.

| Component | Configuration | Time Cost | Ratio |
|---|---|---|---|
| Low-rank tensor prior | t-SVD, max_iter=20 | 5.97 s | 38.7% |
| Latent diffusion sampling | 128 steps | 2.57 s | 16.6% |
| Physical guidance | $t \in [0.1, 0.8]$, freq=2 | 6.85 s | 44.4% |
| VAE encode/decode | Included in above | Negligible | $< 0.1\%$ |
| Total Runtime | Per sample (RTX 4090) | 15.43 s | 100% |

Table 6 presents the computational breakdown of LoPhyDA, showing a total inference time of 15.43s per sample on an NVIDIA RTX 4090. While Physical guidance remains the primary computational bottleneck (44.4%) due to iterative PDE gradient calculations, the Low-rank tensor prior efficiently establishes global structural constraints in just 5.97s. Notably, the Latent diffusion sampling is highly optimized, consuming only 16.6% of the total time for 128 steps, demonstrating the efficiency of performing denoising within the compressed latent space.

## C.3. Parameter analysis

In this section, we investigate the sensitivity of the model to two critical hyperparameters: the physics guidance weight $\rho$ and the singular value thresholding $\mu$.

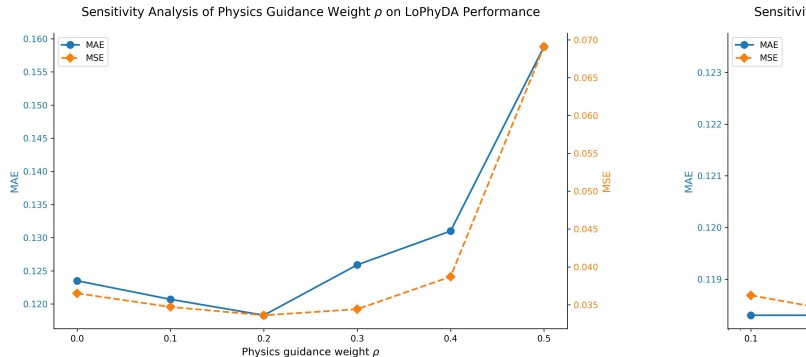

*Figure 4.* Impact of physics guidance weight $\rho$.        *Figure 5.* Impact of singular value thresholding $\mu$

**Physics guidance weight.** This weight balances the physical constraints against data-driven loss. As shown in Fig . 4, an optimal value ensures the model adheres to physical laws without sacrificing its ability to learn complex empirical patterns, preventing the model from being over-constrained by rigid physical prior.

**Singular value thresholding $\mu$.** The parameter $\mu$ serves as a regulator for the low-rank constraint by controlling the threshold $\tau$. As illustrated in Fig. 5, a smaller $\mu$ leads to a larger $\tau$, which filters more singular values to produce a smoother and lower-rank representation. While this promotes denoising, an excessively small $\mu$ may discard critical structural details, necessitating a careful balance between smoothness and information preservation.

## D. Real observation setting

**Dataset description.** The GDAS prepbufr dataset provides a comprehensive suite of global meteorological observations managed by the National Centers for Environmental Prediction (NCEP). It encompasses surface and upper-air measurements collected from diverse sources, including land and marine stations, radiosondes, aircraft reports, and Global Telecommunication System (GTS) transmissions. The dataset also integrates advanced remote sensing inputs, such as wind profilers,

radar-derived wind data from the U.S., satellite-derived winds from NESDIS, and ocean surface wind estimates from instruments like SSM/I. As a rich and operationally relevant benchmark for modern weather forecasting systems, this dataset is ideally suited for evaluating real-world data assimilation performance.

**Data preprocessing.** Each entry in the PREPBUFR dataset represents a single measurement recorded by a specific instrument at a known time and spatial location. Since our assimilation algorithm targets fixed assimilation times, we retain only those observations falling within a narrow time window, specifically ±30 minutes around each assimilation cycle. For instance, for an assimilation time of 6:00 PM, we include only observations recorded between 5:30 PM and 6:30 PM. Furthermore, the spatial locations of these observations are typically irregular and do not align directly with the forecast model grid. To address this, we constructed a new, finer observation grid and projected the raw observations onto it using nearest-neighbor mapping. This intermediate grid matches the horizontal resolution of the FengWu model but contains 40 vertical levels—significantly denser than the original FengWu grid—to preserve vertical structure while maintaining computational efficiency.

## E. More visualization results

We provide more visualization results to further evaluate the performance of LoPhyDA. Figures 4–7 illustrate the single-step assimilation experiments comparing different methods under the 1% observation condition. In these figures, the top row displays the ERA5 ground truth, the background field, and the observation field (from left to right). The middle row presents the assimilation results for DiffDA, AlignDA, and LoPhyDA, respectively, while the bottom row illustrates the absolute error fields relative to the ERA5 ground truth. LoPhyDA consistently exhibits significantly smaller error magnitudes, demonstrating superior error reduction capability compared to other approaches. Furthermore, Figures 8–10 demonstrate the results for other key variables obtained from the 15-day cyclic assimilation experiments.

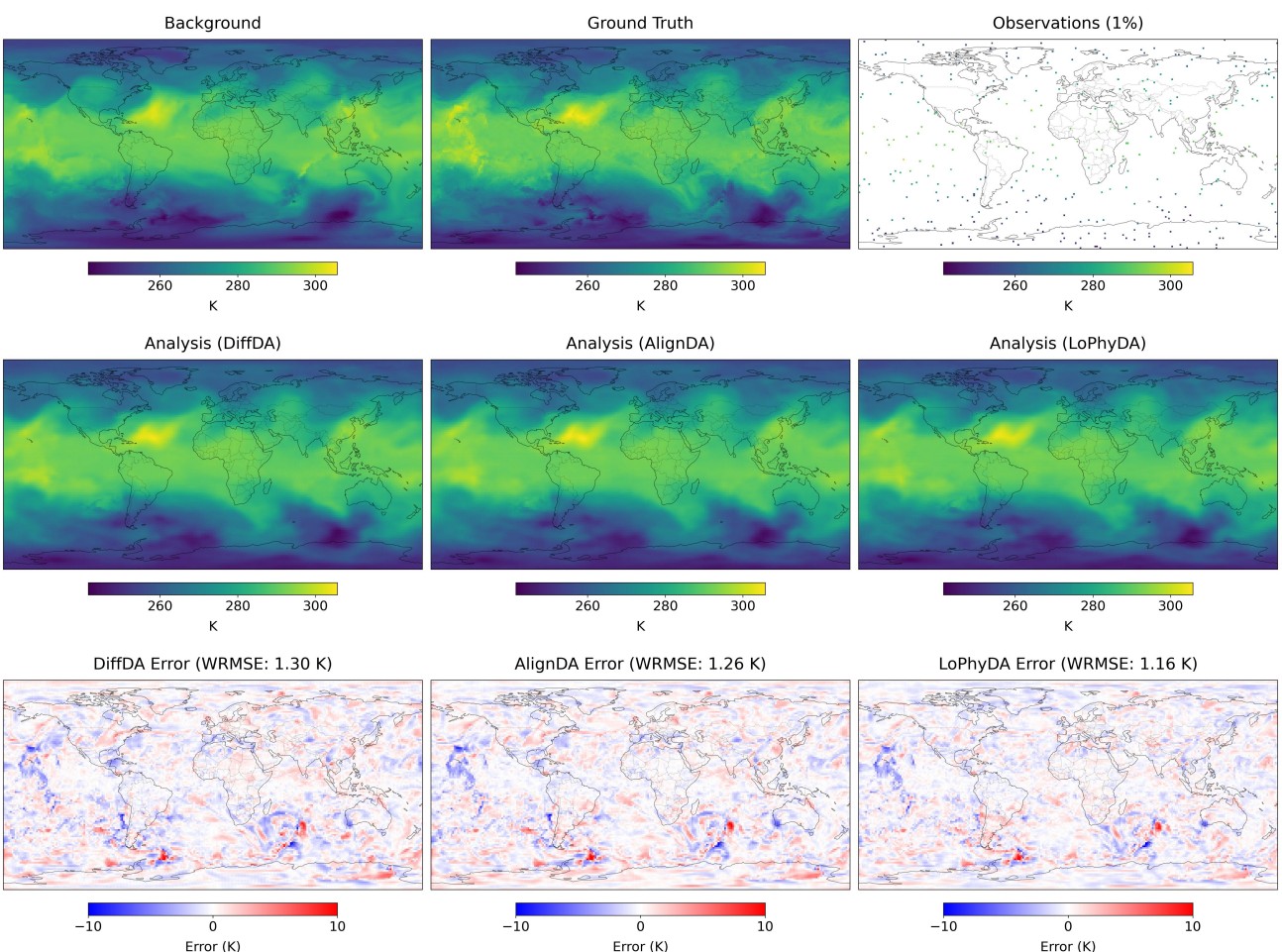

*Figure 6.* Visualization of t850 at a 2019-09-01-18:00 UTC.

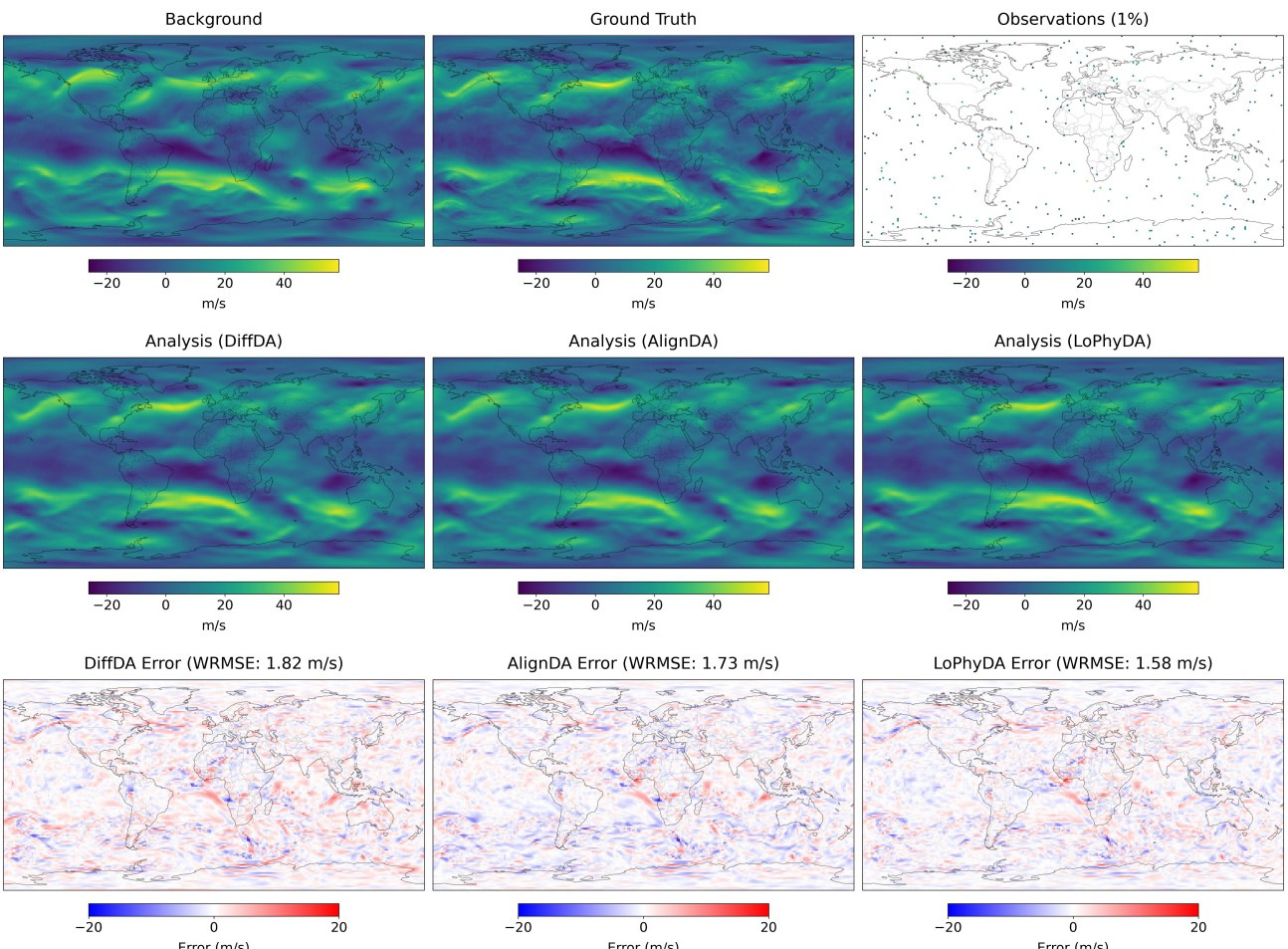

*Figure 7.* Visualization of u500 at a 2019-03-02-6:00 UTC.

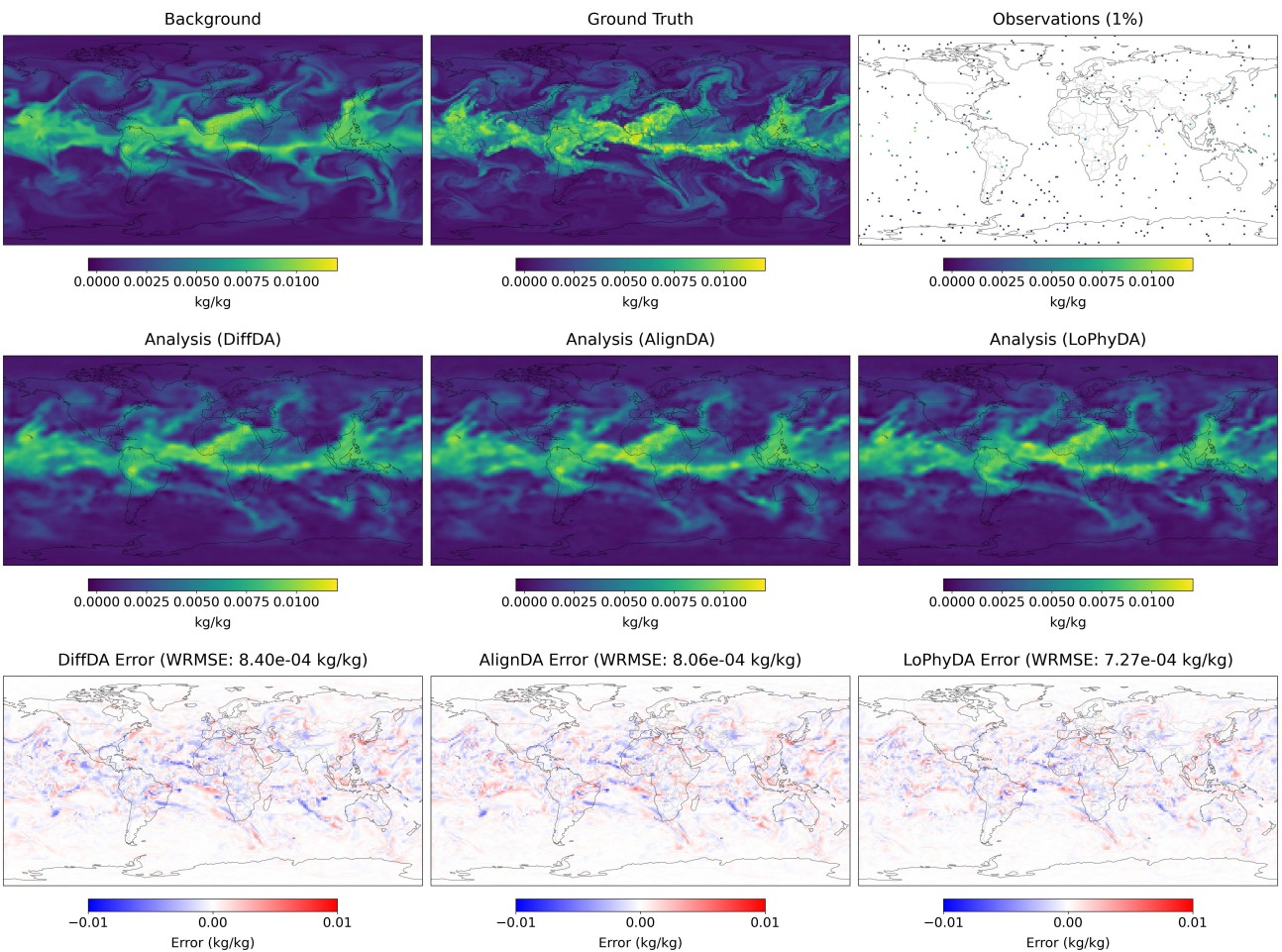

*Figure 8.* Visualization of q700 at a 2019-06-08-18:00 UTC.

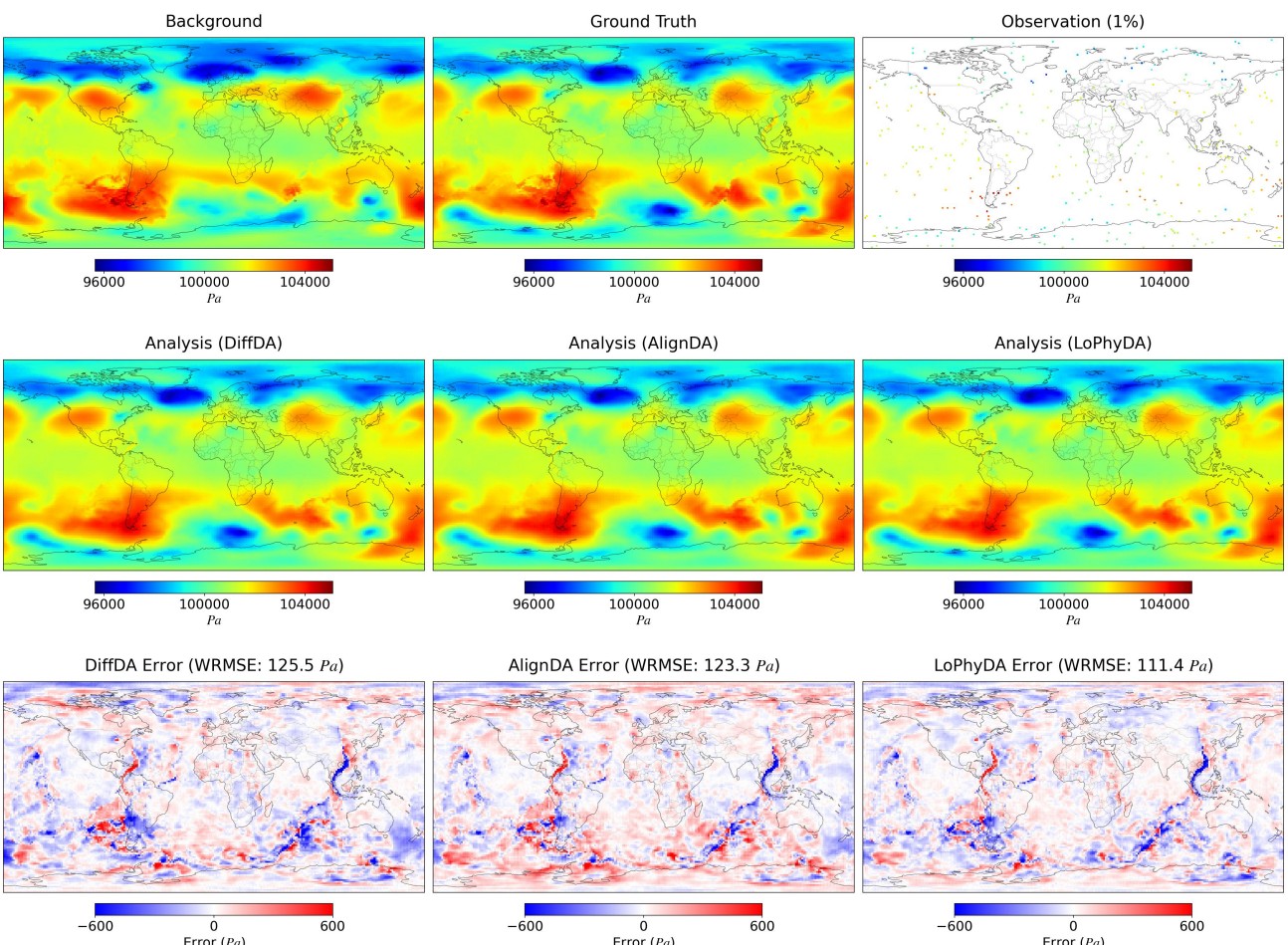

*Figure 9.* Visualization of msl at a 2019-12-18-12:00 UTC.

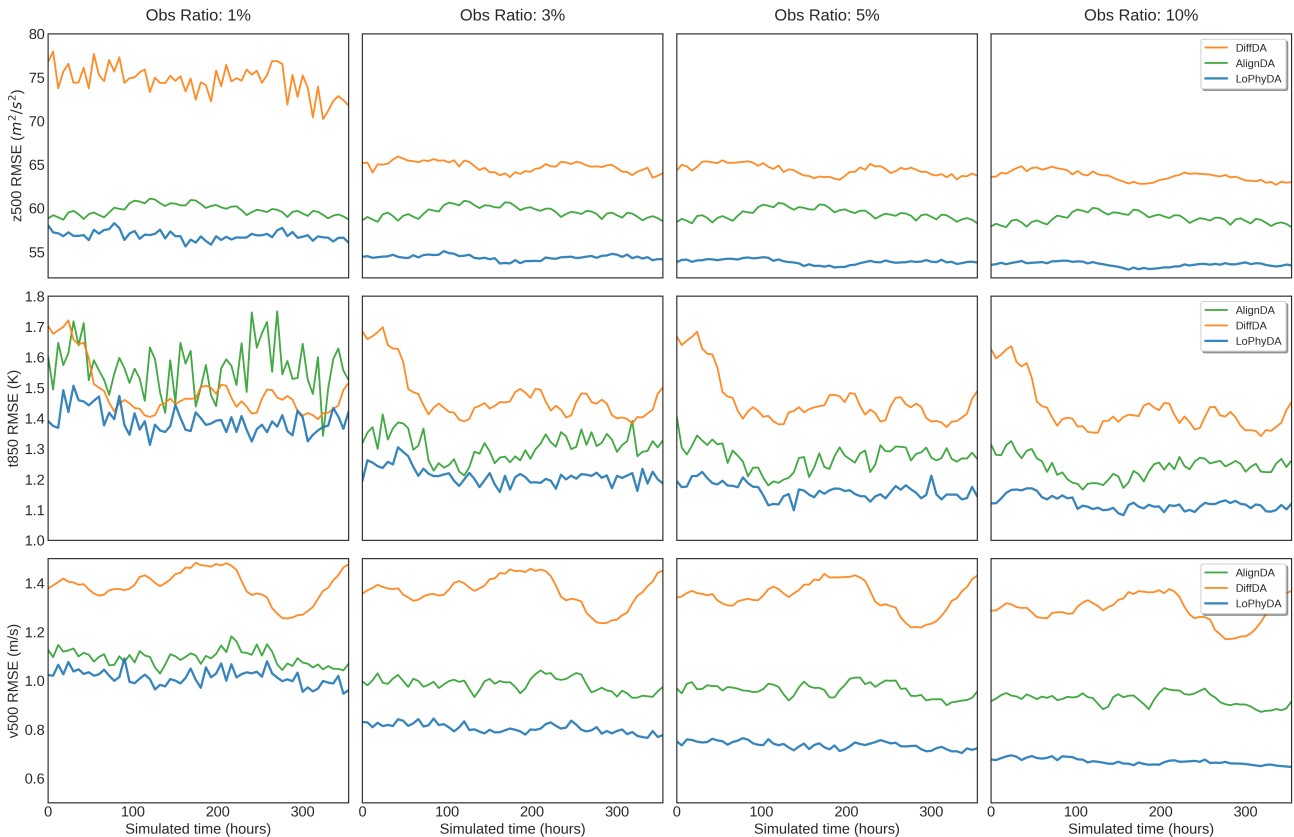

*Figure 10.* Results for unfixed observation positions (15 days). The system is simulated in an auto-regressive manner for 15 days, starting from January 1, 2019. Following AlignDA, we demonstrate the RMSEs of the analysis field for three variables (z500, t850, v500) in three different rows.

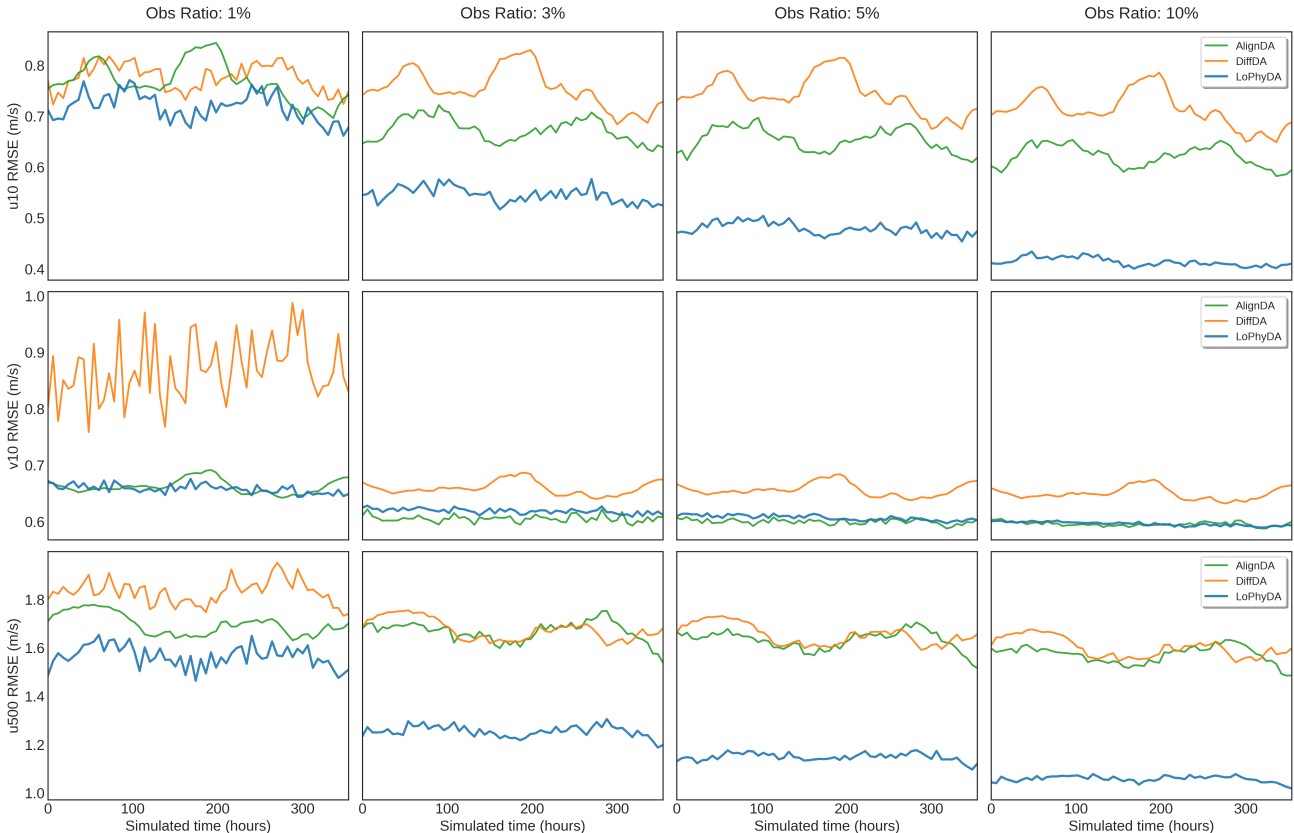

*Figure 11.* Results for unfixed observation positions (15 days). The system is simulated in an auto-regressive manner for 15 days, starting from January 1, 2019. Following AlignDA, we demonstrate the RMSEs of the analysis field for three variables (u10, v10, u500) in three different rows.

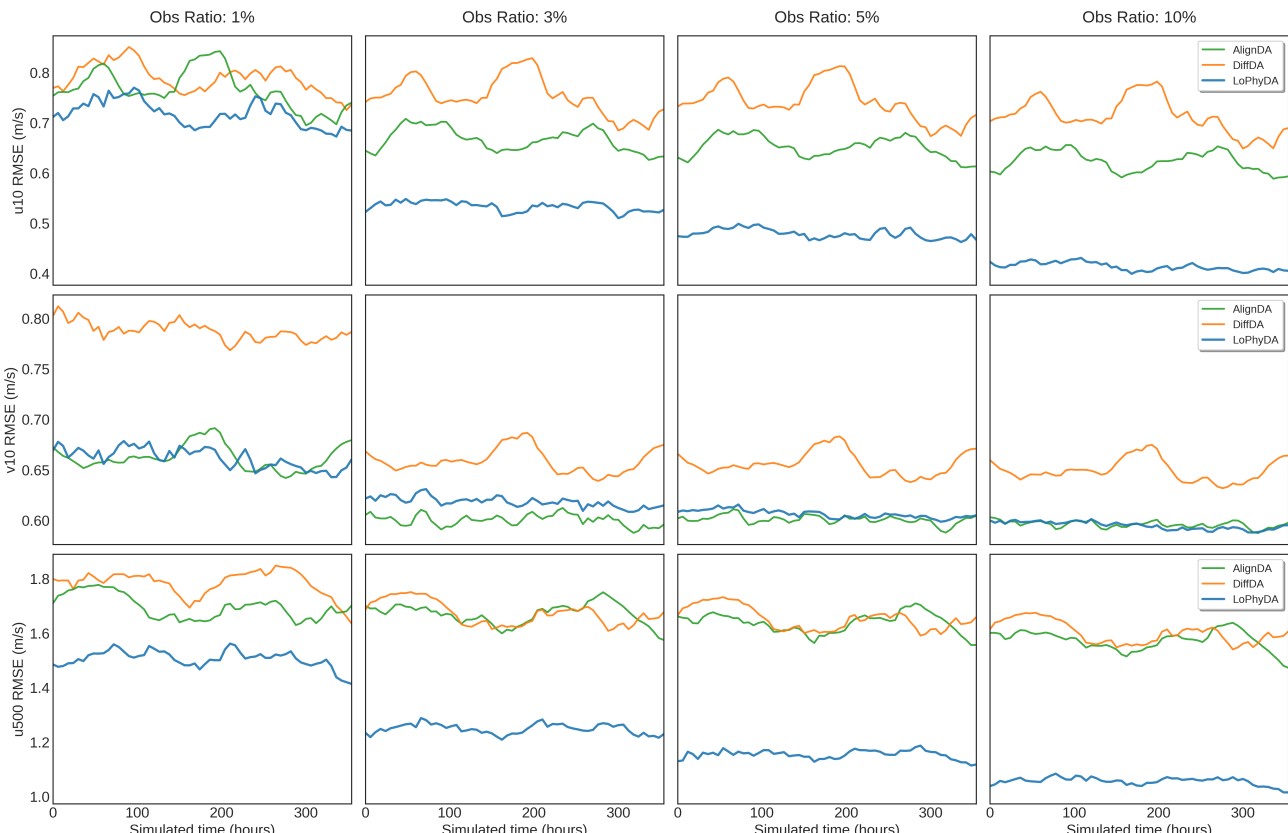

*Figure 12.* Results for fixed observation positions (15 days). The system is simulated in an auto-regressive manner for 15 days, starting from January 1, 2019. Following AlignDA, we demonstrate the RMSEs of the analysis field for three variables (u10, v10, u500) in three different rows.

