# OpenReview forum: "LoPhyDA: Low-Rank Tensor and Physics Gradient Guided Diffusion for Atmospheric Data Assimilation"
_ICML.cc/2026/Conference — ICML 2026 regular_

### Official Review · Reviewer_1hzz · 2026-03-08

**Soundness:** 3
**Presentation:** 3
**Significance:** 3
**Originality:** 3
**Overall Recommendation:** 4
**Confidence:** 4

**Summary:**

This paper tackles the problem of atmospheric data assimilation (DA) under conditions of extreme observational sparsity, a critical challenge for modern weather forecasting. The authors propose LoPhyDA, a novel framework built upon a diffusion model, which introduces a dual-guidance mechanism to reconstruct a complete and physically plausible atmospheric state. By combining a data-driven global prior with a physics-based local correction, LoPhyDA aims to overcome the failure modes of prior diffusion-based DA methods, which tend to produce physically inconsistent or erroneous states in unobserved regions. The authors validate their approach on the large-scale ERA5 dataset, demonstrating quantitative and qualitative improvements over recent state-of-the-art baselines.

**Compliance With Llm Reviewing Policy:**

Affirmed.

**Final Justification:**

All overall, the author addressed my concern, I would like to remain my score as it is.

**Key Questions For Authors:**

1. How does the model perform in tropical regions where the geostrophic balance constraint is invalid? Does the physics guidance term become negligible, or could it potentially introduce erroneous gradients?

2. The low-rank tensor completion provides a powerful global prior. How does the performance of the overall system degrade if a simpler interpolation method (e.g., kriging or optimal interpolation) is used to generate the global prior instead of the computationally more expensive t-SVD approach?

3. Is the physical guidance really necessary? I mean I am not sure if the physical constraints are always true in the weather forcast domain given some assumptions are not accurate. Please comment on this

**Limitations:**

Yes.

**Strengths And Weaknesses:**

Strenths:
1. The paper presents a comprehensive set of experiments that convincingly demonstrate the method's superiority. The comparisons to recent, relevant baselines (DiffDA, AlignDA) are fair, and LoPhyDA shows consistent improvements across multiple metrics, observation densities, and in both simulated and real-world observation settings.

2. Over all, the paper is well-written and easy to follow. It includes excellent qualitative visualizations (Figure 2, and more in the appendix) that allow the reader to visually inspect the quality of the reconstructed fields and error maps. This provides compelling intuitive evidence that the method is working as intended.

Weakness:
1. While the combination of the two guidance mechanisms is novel, the individual components are based on well-established ideas. Low-rank tensor completion has been widely used for spatio-temporal data imputation [1], and physics-guided diffusion is a rapidly growing subfield with many recent contributions [2]. Please comment

[1] G. Si, M. Xie, F. Zhang, T. Xia, L. Xi, "A framework for wind field forecasting from sparse observations via integrated tensor completion and prediction," Expert Systems with Applications, 2026.
[2] H. Wang, J. Han, W. Fan, W. Zhang, H. Liu, "PhyDA: Physics-guided diffusion models for data assimilation in atmospheric systems," arXiv preprint arXiv:2505.12882, 2025.

2. The sole physical guidance used is the geostrophic balance. The analysis would be more complete if it acknowledged the limitations of this specific physical choice and discussed its performance in non-geostrophic regimes. Please comment

---

> ### Author Rebuttal · Authors · 2026-03-30
>
> We sincerely thank the reviewer for the careful reading and constructive suggestions.
> Below we provide clarifications and relevant additional experiments in response to the reviewer’s questions.
> ### **W1**
>
> We have substantially improved each core component of the framework. For the low-rank tensor completion, we integrate meteorological multivariate characteristics to reconstruct data in the Fourier domain. Through tensor slicing, we ensure the physical independence of each variable during reconstruction, while simultaneously leveraging low-rank constraints to achieve cross-variable joint reconstruction.
>
> Regarding physical guidance, while most existing methods impose constraints during the denoising phase of the model's *training* process, our approach fundamentally differs by applying gradient guidance during the denoising phase of the *inference* (testing) process from a Bayesian perspective. Our method is highly flexible: it does not require retraining when facing observation data of varying densities, and it supports the flexible integration of different physical equations, achieving true training-free data assimilation.
>
> ---
>
> ### **W2**
>
> We completely agree with your assessment: as an idealized dynamical approximation, geostrophic balance indeed has inherent limitations. Following your suggestion, we have added a comprehensive discussion on the limitations of this physical choice in the Limitation section of our revised manuscript, and analyzed the model's performance in non-geostrophic regions.
>
> **Limitation:** As an idealized dynamical approximation, the geostrophic balance assumption fails in equatorial and low-latitude regions. In future work, we plan to incorporate alternative physical equations and design specific modules to disable the geostrophic balance constraint in these low-latitude areas, relying instead on other physical equations to ensure physical consistency.
>
> ---
>
> ### **Q1**
>
> Indeed, introducing this prior in non-geostrophic regions (such as the equator) will generate erroneous guidance gradients. However, because our physical guidance acts as a soft constraint, this local deviation does not cause the reconstruction results in these areas to deteriorate significantly. Moreover, from a global perspective, the physical guidance still substantially improves the overall model performance. Naturally, the current inability to completely eliminate this physical bias in equatorial regions is a limitation of this work, which will be a key focus for our future optimization efforts.
>
> ---
>
> ### **Q2**
>
> Thank you very much for your suggestion. The absence of this type of baseline comparison was indeed a shortcoming in our original manuscript. We have now supplemented these experiments as shown in the table below. Specifically, learnable encoders must be retrained for varying observation densities during the training phase, resulting in poor generalizability. Conversely, although interpolation methods are training-free, they fail to capture global structural information, often leading to localized fitting. As the results demonstrate, our low-rank approach successfully overcomes these limitations, yielding a more effective and dense observation field.
>
> | Method | MAE | MSE |
> |:--|:--|:--|
> | Kriging Interpolation | 0.1328 | 0.0391 |
> | Learnable Encoder (PhyDA) | 0.1237 | 0.0352 |
> | **Low-Rank (Ours)** | **0.1183** | **0.0336** |
>
> ---
>
> ### **Q3**
>
> A review of recent literature confirms the necessity of physical guidance in the field of weather forecasting [1, 2]. Regarding the specific physical equation utilized in our method, reference [1] similarly employed geostrophic balance as physical guidance within a reinforcement learning framework.
>
> [1] Luo, Yingtao, et al. "Physics-guided learning of meteorological dynamics for weather downscaling and forecasting." *Proceedings of the 31st ACM SIGKDD Conference on Knowledge Discovery and Data Mining V. 2.* 2025.
>
> [2] Sun, J.-A., "Align-DA: Align score-based atmospheric data assimilation with multiple preferences." In *The Thirty-ninth Annual Conference on Neural Information Processing Systems*, 2025.

---

> > ### Author Rebuttal · Reviewer_1hzz · 2026-04-03
> >
> > Thanks for your rebuttal. My concern has been addressed.

---

> > > ### Author Response · Authors · 2026-04-03
> > >
> > > Thank you for your acknowledgement. We are glad that our explanations and additional experiments have sufficiently addressed your concerns. We sincerely hope these clarifications will further strengthen your confidence in the quality of our work.

---

### Official Review · Reviewer_gP2D · 2026-03-10

**Soundness:** 2
**Presentation:** 1
**Significance:** 2
**Originality:** 3
**Overall Recommendation:** 4
**Confidence:** 3

**Summary:**

This paper introduces LoPhyDA, a data assimilation method designed for sparse observations settings. The idea is to iteratively construct a dense observation tensor from sparse measurements using a tensor completion technique based on singular value decomposition in the Fourier space. The resulting dense tensor is then used to condition a diffusion process that generates an analysis $x$ from a forecast $\hat{x}$. Using posterior sampling techniques, a physics-based guidance term is also incorporated in the diffusion process to enforce dynamical consistency.

**Compliance With Llm Reviewing Policy:**

Affirmed.

**Final Justification:**

The rebuttal addressed my main concern about the clarity and presentation of the paper. Also, in their rebuttals to other reviewers, the authors showed that the proposed low-rank method yields a more effective observation field. Thus, given the overall positive feedback from other reviewers, I decide to increase my score to weak accept.

**Key Questions For Authors:**

1) Could the authors clarify how the background $\hat{x}$ is used during the diffusion process? In Algorithm (1) provided in the appendix, the background $\hat{x}$ is given as input but does not appear to be used in the algorithm.

2) Could the authors confirm that $E$ is a notation for the encoder? In Algorithm (1), $\mathcal{E}$ is described as the encoder, but only $E$ is used in the text.

3) Is $\mu_{\theta} E(y^{'})$ an Hadamard product ? Is so, could the authors elaborate on the idea/intuition that motivated the integration of the dense observation tensor this way?

4) In the data assimilation literature, the problem is often formulated as a Bayesian filtering task, aiming to obtain samples from the filtering distribution $p(x_{t} \mid y_{1:t})$. Could the authors clarify the connection between their recursive algorithm $(x_{0} \rightarrow x_{1}^{b} \rightarrow x_{1}^{a} \rightarrow x_{2}^{b} \rightarrow ...)$ and this distribution?

I would be willing to raise my score if the authors improve the clarity of the manuscript and include SDA in the second experiment (assimilation on complete trajectories), or provide an explanation as to why this would not necessarily be relevant.

**Limitations:**

Yes, the limitations are clearly discussed by the authors in Section 5, which is commendable.

**Strengths And Weaknesses:**

Strenghts:
* The proposed tensor completion approach is original and particularly interesting in the context of meteorology.
* This method has significant practical value because observations from ground weather stations are sparse in space.
* The results obtained in comparison with related methods are promising.
* The use of real observations in the experiments is commendable.

Weaknesses:
* Overall, I find the paper poorly written and difficult to follow. For instance, the algorithm that is actually used in practice is only provided in the appendix, whereas it would be more appropriate to present it in the main text. In addition, several notations are introduced without definitions. For example, what do $\alpha_{t}$ in Equation (5), $\rho$ in Equation (7), and $(\mu_{t}/E/\sigma_{t})$ on line 255 refer to ?
* In the first experiment, which involves single-step assimilation (i.e., with only one observation $y_{t}$), LoPhyDA is compared with SDA [1], a Bayesian smoothing approach originally designed to reconstruct complete trajectories from multiple observations. Therefore, a comparison with SDA would have been more appropriate in the second experiment, which aims to estimate a full trajectory. However, SDA is not evaluated in that case.

[1] Rozet et al., Score-based Data Assimilation, Thirty-seventh Conference on Neural Information Processing Systems, 2023.

---

> ### Author Rebuttal · Authors · 2026-03-30
>
> We sincerely thank the reviewer for the careful reading and constructive suggestions.
> Below we provide clarifications and relevant additional experiments in response to the reviewer’s questions.
> ### **W1**
>
> We have moved Algorithm 1 from the appendix to the main text and completed the missing definitions following Equation (5): Here, $\nabla_{z_t}\log p(z_t)$ denotes the score of the data distribution, and $\alpha_t = \prod_{s=1}^t (1-\beta_s)$ represents the cumulative noise schedule parameter at step $t$, which is derived from the predefined variance schedule $\beta_s$.
>
> ---
>
> ### **W2**
>
> We have supplemented the comparison data with SDA in the second set of experiments (please refer to the SDA cyclic assimilation comparison result graph:
> [https://anonymous.4open.science/r/LoPhyDA-642F/sda_recycle.jpg](https://anonymous.4open.science/r/LoPhyDA-642F/sda_recycle.jpg)).
>
> The experimental results demonstrate that during the cyclic assimilation process, our algorithm achieves a more significant error reduction compared to SDA.
>
> ---
>
> ### **Q1**
>
> We utilize the background field as a condition for the diffusion model and have corrected the description of the noise prediction in Line 7 of Algorithm 1:
>
> $$
> \epsilon \leftarrow \epsilon_{\theta}(z_{t}, t, E(\hat{x}))
> $$
>
> *(Conditional noise prediction with background $\hat{x}$).*
>
> ---
>
> ### **Q2**
>
> We have corrected the typographical error regarding the encoder in Algorithm 1, changing it to $E(\cdot)$.
>
> ---
>
> ### **Q3**
>
> We have corrected the description of the fusion process. The updated workflow of Algorithm 1 is as follows (the complete Algorithm is available at:
> [https://anonymous.4open.science/r/LoPhyDA-642F/Overall%20Algorithm.jpg](https://anonymous.4open.science/r/LoPhyDA-642F/Overall%20Algorithm.jpg)), and corresponding revisions have been made in the main text.
>
> ---
>
> ## **Updated Algorithm 1**
>
> **For** $t = N, N-1, \ldots, 1$ **do**
>
> ### Compute cumulative noise level
>
> $$
> \alpha_t = \prod_{s=1}^{t}(1-\beta_s)
> $$
>
> ### **1. Physics gradient guidance**
>
> **Conditional noise prediction with background $\hat{x}$:**
> $$
> \epsilon \leftarrow \epsilon_{\theta}(z_t, t, E(\hat{x}))
> $$
>
> **Estimate clean latent via Tweedie's formula:**
> $$
> \hat{z}_0 \leftarrow \frac{1}{\sqrt{\alpha_t}}\left(z_t - \sqrt{1-\alpha_t},\epsilon\right)
> $$
>
> **Decode to physical space for PDE check:**
> $$
> \hat{x}_0 \leftarrow \mathcal{D}(\hat{z}_0)
> $$
>
> **Backpropagate physics loss:**
> $$
> g \leftarrow \nabla_{z_t}\mathcal{L}_{\mathrm{physics}}(\hat{x}_0)
> $$
>
> **Inject physics gradient:**
> $$
> \hat{\epsilon} \leftarrow \epsilon + \rho\sqrt{1-\alpha_t},g
> $$
>
> ### **2. Predict state**
>
> **Compute posterior mean:**
> $$
> \mu_{\theta} = \frac{1}{\sqrt{1-\beta_t}}\left(z_t - \frac{\beta_t}{\sqrt{1-\alpha_t}}\hat{\epsilon}\right)
> $$
>
> **Predict next state:**
> $$
> \tilde z_{t-1} \sim \mathcal{N}(\mu_\theta,; \frac{1-\alpha_{t-1}}{1-\alpha_t}\beta_t I)
> $$
>
> **Sample noisy reconstructed observation at $t-1$:**
> $$
> z_{t-1}^{\mathrm{rec}} \sim \mathcal{N}\left(\sqrt{\alpha_{t-1}},E(y^{\prime}), (1-\alpha_{t-1})I\right)
> $$
>
> **Fuse via Gaspari--Cohn:**
> $$
> z_{t-1} \leftarrow E\left(L \odot D(z_{t-1}^{\mathrm{rec}}) + (1-L) \odot D(\tilde{z}_{t-1})\right)
> $$
>
> **End for**
>
> Although the low-rank tensor completion provides a global prior, its reconstruction accuracy is highest near the actual observation points and decays with distance. Therefore, we utilize the function $L$ to implement dynamic weight allocation via the Hadamard product ($\odot$): the weight is high near real observation points, while it is reduced in regions with sparse or missing observations. This soft constraint enables the efficient assimilation of reconstructed observation data while simultaneously mitigating the errors introduced by reconstructing unreliable observation regions.
>
> ---
>
> ### **Q4**
>
> The recursive algorithm ($x_0 \rightarrow x_1^b \rightarrow x_1^a \dots$) is a mapping of the forecast-analysis cycle from Bayesian filtering into the diffusion model framework, aiming to sample from the filtering distribution $p(x_t \mid y_{1:t})$.
>
> * **Forecast step** ($x_{t-1}^a \rightarrow x_t^b$):
>   The model evolves based on the analysis state from the previous time step, implicitly sampling from the prior distribution $p(x_t \mid y_{1:t-1})$.
>
> * **Analysis step** ($x_t^b \rightarrow x_t^a$):
>   Based on the background state $x_t^b$, the model introduces the low-rank tensor constraint to optimize the likelihood gradient of the observation data, $\nabla_x \log p(y_t \mid x_t)$. Simultaneously, it incorporates the physical gradient $\nabla_x \log p_{phys}(x_t)$ for conditional denoising, thereby solving for the target posterior distribution.
>
> This alternating progression of $x_t^b$ and $x_t^a$ is strictly equivalent to the Bayesian recursive inference process of continuously updating the system state with new observations.

---

> > ### Author Rebuttal · Reviewer_gP2D · 2026-04-03
> >
> > I thank the authors for improving the structure and clarity of the paper and for adding SDA to the second experiment. I also appreciate the response to the second question from reviewer 5S4J, which clarifies the value of the proposed low-rank method. For these reasons, I have decided to increase my score to weak accept.

---

> > > ### Author Response · Authors · 2026-04-03
> > >
> > > Thank you for your careful review of our manuscript and rebuttal. We are pleased that the clarifications and additional experiments have  sufficiently addressed your concerns. We highly appreciate your reconsideration and updated rating of our manuscript.

---

### Official Review · Reviewer_BSUa · 2026-03-10

**Soundness:** 3
**Presentation:** 2
**Significance:** 3
**Originality:** 3
**Overall Recommendation:** 4
**Confidence:** 5

**Summary:**

This paper seeks to encode in low-rank reconstructed observation field as a conditioning mechanism for generation of the assimilated posterior in the latent space. They use physical constraints to further enhance the model's performance, evaluating on ERA5 with synthetic observations and some real world observations too. They achieved improved performance over methods such as DiffDA, SDA, and AlignDA.

**Compliance With Llm Reviewing Policy:**

Affirmed.

**Final Justification:**

Rebuttal addressed my main concern, raising score to a 4.

**Key Questions For Authors:**

1. See weaknesses (above), this is my main concern for the authors. I would be willing to raise my score if this is addressed well.
2. If LoPhyDA physics guidance were incorporated with DiffDA conditioning, or the Low-Rank Tensor conditioning were incorporated with AlignDA constraints, how well would the model do?

**Limitations:**

Yes.

**Strengths And Weaknesses:**

Strengths:

Results seem strong, compared to baselines of SDA, DiffDA, and AlignDA, and the authors ensure consistency of model architecture here.

Ablation studies demonstrate the effectiveness of each of the components, and the improving performance with less sparsity results make sense.

Low rank tensor reconstruction of the observation representation seems to be new.

Weaknesses:

I'm reading [1] and the method seems to be fairly similar with the gradient guidance, though the authors claim that their method approaches this part differently from the perspective of posterior sampling. In particular, the results defined in equation 8 in this paper is very similar to the results of equation 6 in Theorem 1 from [1], with only some minor notational differences from my point of view. Can the authors further discuss what differentiates their method from the existing paper, other than the minor architectural differences in the low rank tensor prior? If there is no difference, I do have some additional concerns about the novelty of this portion of the method.

[1] Wang, H., Han, J., Fan, W., Zhang, W., and Liu, H. Phyda:
Physics-guided diffusion models for data assimilation in
atmospheric systems. arXiv preprint arXiv:2505.12882,
2025a.

---

> ### Author Rebuttal · Authors · 2026-03-30
>
> We sincerely thank the reviewer for the careful reading and constructive suggestions.
> Below we provide clarifications and relevant additional experiments in response to the reviewer’s questions.
> ## **Q1**
>
> **Distinctions in Physical Guidance：**
>
> PhyDA is an end-to-end training approach. Equation (6) in PhyDA introduces a physical regularization training objective by incorporating physical residuals into the score-matching training process. Consequently, the neural network is forced to fit both the data distribution and the PDE residuals during training. Therefore, PhyDA necessitates modifying the diffusion model's loss function and retraining the model from scratch to learn physical laws.
>
> Conversely, LoPhyDA is a training-free, inference-time guidance method. Our Equation (8) is not a training objective but is applied during the testing (inference) phase. LoPhyDA does not require retraining the diffusion model. Instead, during each step of the reverse diffusion sampling, we compute the physical residual gradient and inject it directly into the predicted noise $\epsilon$ as a guidance signal. This grants LoPhyDA its highly flexible, plug-and-play characteristics. Furthermore, the physical weight $\rho$ is an inference-time hyperparameter that can be dynamically adjusted for different assimilation tasks, and the PDEs can even be flexibly swapped during inference without retraining any network parameters.
>
> $$
> \epsilon_{\phi}^{\prime} = \epsilon_\phi(z_t, t) + \rho \sqrt{1 - \alpha_t} \nabla_{z_t} \mathcal{L}_{\mathrm{physics}}
> $$
>
> As illustrated in this formula, $\epsilon_\phi$ is the noise predicted by the pre-trained denoising model. By adding the physical gradient to this noise, we obtain the adjusted, physics-informed noise $\epsilon'_\phi$.
>
> **Distinctions in Handling Sparse Observations：**
>
> PhyDA forces the neural network to directly learn the coupling relationships between variables from sparse observations. When observations are extremely sparse, it is prone to establishing spurious (unreasonable) correlations between different variables. Moreover, as a purely data-driven approach, it exhibits poor generalization performance when encountering unseen levels of extreme sparsity.
>
> In contrast, LoPhyDA employs a decouple first, jointly reconstruct later strategy. We first decouple the variable relationships in the Fourier domain and then achieve cross-variable joint reconstruction through a low-rank constraint along the variable dimension. This strategy ensures the relative physical independence of each variable's reconstruction while simultaneously capturing cross-variable correlations through a shared low-rank structure. Importantly, this is a numerical optimization process solved iteratively online during assimilation. Because it requires no training, it demonstrates strong robustness to irregular data and varying degrees of observation sparsity.
>
> **Experimental Comparison：**
>
> We have supplemented the baseline experiments with PhyDA, as shown in the table below. LoPhyDA achieves significant reductions of **3.55%** and **3.27%** in overall MAE and MSE compared to PhyDA, respectively, fully validating the superiority of our dual-guidance approach utilizing both the low-rank global prior and physical gradients.
>
> | Model              | MAE                 | MSE                 | z500    | t850   | t2m    | u10    | v10    | u500   | v500   |
> | :----------------- | :------------------ | :------------------ | :------ | :----- | :----- | :----- | :----- | :----- | :----- |
> | PhyDA              | 0.1225              | 0.0347              | 71.6741 | 1.1033 | 4.7928 | 0.8062 | 0.7196 | 1.2435 | 1.4766 |
> | **LoPhyDA (Ours)** | **0.1183** (↓3.55%) | **0.0336** (↓3.27%) | 58.8376 | 1.1217 | 4.4293 | 0.7178 | 0.6811 | 1.4919 | 1.0229 |
>
> ---
>
> ## **Q2**
>
> As demonstrated by the experimental results below, seamlessly integrating our physical guidance strategy into DiffDA, and incorporating our low-rank tensor constraint into AlignDA, both yield consistently lower MAE and MSE compared to their original baseline models. This compellingly proves that our designed modules are not merely architecture-specific fine-tunes. Rather, they are highly generalizable, plug-and-play modules capable of consistently providing performance enhancements across broader diffusion-based data assimilation frameworks.
>
> | Method                                              | MAE                         | MSE                         |
> | :-------------------------------------------------- | :-------------------------- | :-------------------------- |
> | DiffDA $\rightarrow$ DiffDA + Physical Guidance     | 0.1368 $\rightarrow$ 0.1346 | 0.0375 $\rightarrow$ 0.0371 |
> | AlignDA $\rightarrow$ AlignDA + Low-Rank Constraint | 0.1267 $\rightarrow$ 0.1197 | 0.0353 $\rightarrow$ 0.0342 |
> | **LoPhyDA (Ours)**                                  | **0.1183**                  | **0.0336**                  |

---

> > ### Author Rebuttal · Reviewer_BSUa · 2026-04-02
> >
> > See comment.

---

> > > ### Author Response · Authors · 2026-04-02
> > >
> > > Thank you for your acknowledgement. We are pleased that the clarifications and additional experiments have proven helpful in addressing your concerns. We highly appreciate your reconsideration and updated rating of our manuscript.

---

### Official Review · Reviewer_5S4J · 2026-03-12

**Soundness:** 2
**Presentation:** 3
**Significance:** 2
**Originality:** 2
**Overall Recommendation:** 4
**Confidence:** 3

**Summary:**

This paper studies atmospheric data assimilation under sparse observations. The authors propose LoPhyDA, a diffusion-based DA framework with two guidance mechanisms. First, sparse observations are reconstructed into a dense global prior through low-rank tensor completion based on t-SVD and ADMM. Second, during reverse diffusion, the model injects physics gradients derived from a geostrophic-balance residual to steer sampling toward physically plausible states. The paper evaluates the method on ERA5 with simulated sparse observations and on GDAS PrepBUFR real observations, and also includes 15-day cyclic assimilation experiments, component ablations, observation-density sensitivity, and a runtime breakdown. The reported results show consistent improvements over the diffusion-based baselines considered in the paper, especially in sparse-observation settings.

**Compliance With Llm Reviewing Policy:**

Affirmed.

**Final Justification:**

The authors addressed majority of my comments.

**Key Questions For Authors:**

1. The paper cites PhyDA and Appa, both of which are very close in spirit to the present submission. Can the authors clarify the substantive conceptual difference from these works, and ideally provide a direct empirical comparison at least to PhyDA? A strong answer here would materially improve my view of the paper's originality.


2. How much of the gain comes specifically from the low-rank tensor prior, as opposed to simply providing any dense proxy observation field? Please compare against simpler or learned alternatives for observation reconstruction (e.g., interpolation, smoothing, or a learned encoder-style mapper). If LoPhyDA still wins clearly, that would strengthen the soundness of the low-rank claim.

**Limitations:**

The paper does acknowledge runtime limitations and the mismatch between observation variables and background variables, which I appreciate. However, the limitations discussion should also explicitly cover the regime dependence of geostrophic balance and the possible effect of nearest-neighbor preprocessing for real observations.

**Strengths And Weaknesses:**

This paper has several real strengths. First, it targets an important problem: atmospheric DA under sparse and spatially discontinuous observations. That is a practically relevant setting, and the motivation is clear. Second, the proposed combination is intuitive: the low-rank tensor prior is meant to address the lack of global constraints in unobserved regions, while physics-gradient guidance is meant to improve local physical plausibility during sampling. Third, the empirical section is broader than average for this line of work: the authors evaluate simulated and real observations, include a 15-day cyclic setting, provide a component ablation, analyze observation density, and report a runtime breakdown rather than ignoring computational cost. The empirical gains over the paper's chosen baselines are also consistent.


My main concern is novelty and positioning with respect to closely related work. PhyDA mentioned in the related work is highly relevant to this work because it also combines physics-guided diffusion with a module specifically designed to bridge observational sparsity. The current baseline scope is understandable given the shared latent diffusion setup, but the paper should explicitly justify why the evaluation is restricted to SDA/DiffDA/AlignDA and discuss the relation to closer concurrent methods, especially PhyDA, and to a lesser extent Appa. Ideally, the authors should provide a direct empirical comparison at least to PhyDA.


Soundness: On soundness, I think the paper is promising but not fully closed. The central empirical claim that the proposed combination improves analysis quality over the included diffusion baselines is supported by the tables and visualizations. However, the evidence for the low-rank component is still incomplete, because the paper only compares against removing that component, not against simpler reconstruction baselines such as interpolation, smoothing, or a learned reconstruction/observation encoder. As written, it is hard to tell whether the gains come from "low-rankness" specifically or more generally from providing any dense proxy observation field.


Presentation: The high-level story is understandable, but the paper contains some inconsistency that reduces confidence. Section 4.2 says the background is initialized from Dec. 30, 2018 and evolved for 48 hours, while Table 1 is described as averaged over the entire 2019 test set, which needs clarification.


I appreciated that the paper includes an overhead analysis. Appendix D reports a total inference cost of about 15.43 seconds per sample on an RTX 4090, with physical guidance being the main bottleneck. That is useful information.

---

> ### Author Rebuttal · Authors · 2026-03-30
>
> We sincerely thank the reviewer for the careful reading and constructive suggestions.
> Below we provide clarifications and relevant additional experiments in response to the reviewer’s questions.
> ### **W1**
> The initialization on December 30, 2018, and the subsequent 48-hour evolution (running 8 steps of the FengWu forecast) were performed *solely* to generate the initial background field for January 1, 2019, 00:00 (following the experimental setup in [1]). Using this as the starting point, our data assimilation cycle experiments spanned the entire year of 2019. Therefore, Table 1 correctly presents the average results evaluated on the 2019 test set.
>
> [1] Xiao, Yi, et al. "LoRA-EnVar: Parameter-Efficient Hybrid Ensemble Variational Assimilation for Weather Forecasting." *The Thirty-ninth Annual Conference on Neural Information Processing Systems.*
>
> ---
>
> ### **Q1**
>
> #### **Experimental Comparison with PhyDA**
>
> We have supplemented the comparative experiments with PhyDA, as shown in the table below. LoPhyDA achieves significant reductions of **3.55%** and **3.27%** in overall MAE and MSE compared to PhyDA, respectively. This fully validates the superiority of our dual-guidance approach utilizing both the low-rank global prior and physical gradients.
>
> | Model | MAE | MSE | z500 | t850 | t2m | u10 | v10 | u500 | v500 |
> |:--|:--|:--|:--|:--|:--|:--|:--|:--|:--|
> | PhyDA | 0.1225 | 0.0347 | 71.6741 | 1.1033 | 4.7928 | 0.8062 | 0.7196 | 1.2435 | 1.4766 |
> | **LoPhyDA (Ours)** | **0.1183** (↓3.55%) | **0.0336** (↓3.27%) | **58.8376** | 1.1217 | **4.4293** | **0.7178** | **0.6811** | 1.4919 | **1.0229** |
>
> #### **Distinctions in Physical Guidance**
>
> While both PhyDA and LoPhyDA incorporate physical guidance within the diffusion denoising process, they operate at fundamentally different levels. PhyDA introduces physical constraints as a regularization term in the objective function during the *training* phase to update the denoising network parameters. Consequently, the physical constraints are hardcoded into the model weights, resulting in high retraining costs. In contrast, LoPhyDA does not alter the pre-trained network. Instead, during the *inference* phase, it directly injects the physical residual gradients into each step of the reverse denoising process. Its target is the sampling trajectory of the current sample rather than the model parameters. Therefore, LoPhyDA is a training-free method and offers the advantage of dynamically adjusting the guidance strength based on observation density and time steps.
>
> $$
> \epsilon_{\phi}^{\prime} = \epsilon_\phi(z_t, t) + \rho \sqrt{1 - \alpha_t} \nabla_{z_t} \mathcal{L}_{\mathrm{physics}}
> $$
>
> As illustrated in this formula, $\epsilon_\phi$ is the noise predicted by the pre-trained denoising model. By adding the physical gradient to this noise, we obtain the adjusted, physics-informed noise $\epsilon'_\phi$.
>
> #### **Distinctions in Handling Sparse Observations**
>
> PhyDA forces the neural network to directly learn the coupling relationships between variables from sparse observations. When observations are extremely sparse, it is prone to establishing unreasonable correlations between different variables. LoPhyDA, however, first decouples the variable relationships in the Fourier domain, and then achieves cross-variable joint reconstruction through low-rank constraints along the variable dimension. This strategy ensures the relative physical independence of each variable's reconstruction while simultaneously capturing cross-variable connections through a shared low-rank structure, effectively realizing a "decouple first, jointly reconstruct later under structural constraints" method.
>
> #### **Distinctions from Appa**
>
> Appa designs a spherical geometry image encoder capable of processing global weather data, which differs from conventional planar grid encoders. Its primary function is to achieve the compression of high-dimensional data while preserving spherical features. This objective is fundamentally different to that of LoPhyDA, which aims to reconstruct highly sparse and discrete observation data into observation fields with dense information.
>
> ---
>
> ### **Q2**
>
> We have supplemented the experiments as shown in the table below. It can be observed that compared to Kriging interpolation and the learnable encoder (the method utilized in PhyDA), our low-rank approach yields a more effective and dense observation field. Specifically, learnable encoders must be trained multiple times during the training phase to accommodate different observation densities, which lacks strong generalizability. On the other hand, although Kriging interpolation is a training-free approach, it loses global structural information, resulting in localized fitting.
>
> | Method | MAE | MSE |
> |:--|:--|:--|
> | Kriging Interpolation | 0.1328 | 0.0391 |
> | Learnable Encoder (PhyDA) | 0.1237 | 0.0352 |
> | **Low-Rank (Ours)** | **0.1183** | **0.0336** |

---

> > ### Author Rebuttal · Reviewer_5S4J · 2026-04-02
> >
> > I thank the authors for their detailed rebuttal. The newly added experiments comparing with PhyDA are convincing and address my earlier concern.

---

> > > ### Author Response · Authors · 2026-04-03
> > >
> > > Thank you for your careful review of our manuscript and rebuttal. We are glad that the additional experiments and clarifications have sufficiently addressed your concerns. In light of this, we would highly appreciate your reconsideration of our paper’s contributions and overall evaluation.

---

### Decision · Program_Chairs · 2026-04-30

**Decision:**

Accept (regular)

**Comment:**

The paper proposes a novel atmospheric data assimilation approach based on diffusion models with dual guidance: (a) low-rank tensor priors and (b) physics-based gradients. Reviewers positively highlighted both the relevance and importance of the problem and the extensive empirical validation of the proposed method. The main concerns raised during the review process related to the paper’s novelty and its positioning with respect to prior work.

In the rebuttal, the authors clarified the distinctions between their method and prior approaches such as PhyDA and APPA, and provided additional experimental comparisons that addressed most of the reviewers’ concerns. Overall, the paper presents an interesting and empirically well-supported approach that combines tensor priors with diffusion guidance informed by physics gradients. Given its potential to inspire further progress on an important problem, I recommend acceptance.